# FEATURE DROPOUT: REVISITING THE ROLE OF AUGMENTATIONS IN CONTRASTIVE LEARNING

## ABSTRACT

What role do augmentations play in contrastive learning? Recent work suggests that good augmentations are *label-preserving* with respect to a specific downstream task. We complicate this picture by showing that label-destroying augmentations can be useful in the foundation model setting, where the goal is to learn diverse, general-purpose representations for *multiple* downstream tasks. We perform contrastive learning experiments on a range of image and audio datasets with multiple downstream tasks (e.g. for digits superimposed on photographs, predicting the class of one vs. the other). We find that Viewmaker Networks, a recently proposed model for learning augmentations for contrastive learning, produce label-destroying augmentations that stochastically destroy features needed for different downstream tasks. These augmentations are interpretable (e.g. altering shapes, digits, or letters added to images) and surprisingly often result in better performance compared to expert-designed augmentations, despite not preserving label information. To support our empirical results, we theoretically analyze a simple contrastive learning setting with a linear model. In this setting, label-destroying augmentations are crucial for preventing one set of features from suppressing the learning of features useful for another downstream task. Our results highlight the need for analyzing the interaction between *multiple* downstream tasks when trying to explain the success of foundation models.

## 1 INTRODUCTION

In recent years, foundation models (Bommasani et al., 2021) have exhibited remarkable progress on a range of AI tasks (Devlin et al., 2019; Liu et al., 2019; Ramesh et al., 2021; Radford et al., 2021; Brown et al., 2020; Chowdhery et al., 2022; Hoffmann et al., 2022; Alayrac et al., 2022; Reed et al., 2022). A crucial characteristic of foundation models is that they can be adapted for a range of downstream tasks. For example, a foundation model trained on ImageNet should ideally not only perform well at object classification, but should also have learned general features useful for localization, segmentation, and other visual tasks. Indeed, this is borne out by recent work showing the high accuracy of foundation models on a range of downstream tasks (Chen et al., 2020b), as well as a range of analysis work showing models learn high-level semantic features including texture, color, pose, and style (Goh et al., 2021).

One popular strategy for training foundation models involves training models to match transformed versions (known as *views* or *augmentations*) of the same input. For example, image views might include common data augmentations such as cropping or color jitter (Chen et al., 2020b), while views for speech might include pitch modulation or spectrogram masking (Kharitonov et al., 2021; Park et al., 2019). This family of objectives includes contrastive approaches such as SimCLR and MoCo, as well as non-contrastive approaches such as BYOL and SwAV (Chen et al., 2020b; He et al., 2020; Grill et al., 2020; Caron et al., 2020).

Given the central importance of these views for defining the self-supervised task, much work has focused on the question of what views lead to high-quality representations. The prevailing consensus, exemplified by

(Tian et al., 2020), holds that views should be *label-preserving* with respect to a downstream task. In other words, because the contrastive loss will produce representations which are *invariant* to features that vary across views, any information we wish to preserve in the representations should not be altered by such views. As Tian et al. (2020) write: "*A good set of views are those that share the minimal information necessary to perform well at the downstream task.*"

Here, we question whether this assumption—in particular, with its focus on a single task—is enough to explain why contrastive foundation models succeed on a *range* of downstream tasks. In Section 2, we observe that the actual choice and application of **views in practice** does not align with this prevailing consensus. For example, complete invariance to several common data augmentations (e.g. shifts in brightness or cropping) is undesirable since augmentations of inputs from different classes can collide. Furthermore, in many cases there are explicit ways to specify invariances (e.g. converting images to grayscale) that researchers avoid in favor of specifying them indirectly via augmentations (e.g. hue shifts). These observations suggest that specifying invariances is not the sole role of these views.

Instead, we suspect that augmentations serve as a form of **feature dropout**—preventing any one feature from becoming a shortcut feature and suppressing the learning of other features. We study this idea empirically in Viewmaker Networks, a recently proposed method that appears to learn to drop out different features in the input via adversarial training. We apply viewmaker and expert views to datasets with two associated downstream tasks, one involving classifying the main input (e.g., an image or audio recording) and one involving a simple overlaid element (e.g., a digit, shape, letter, or speech snippet). We observe that the viewmaker augmentations selectively obscure these overlaid features. Despite this, the viewmaker representations still learn both downstream tasks well, while expert views often struggle on one or the other. This further suggests that being label-preserving is not a necessary property of good views, as long as the label information is still *sometimes* accessible.

Finally, we formalize the intuition that feature dropout can aid learning with a theoretical analysis of a simple linear contrastive setting. In this setting, we characterize how the noisiness of each feature directly determines how quickly features are learned, and uncover an **interaction between features** governing how fast they are learned. In particular, we show how learning one feature quickly can suppress the learning of other features, and show that adding noise to the "easiest" feature can *increase* the rate at which other features are learned. This further indicates that *label-destroying* augmentations may have a direct role in ensuring that contrastive models learn a broad range of features for downstream tasks.

Overall, these findings suggest the need to revisit common assumptions about the role of augmentations for contrastive learning in the foundation model setting, and move towards a better understanding of how to train generalist models that learn diverse features from unlabeled data.

## 2 COMMON PRACTICES ARE AT ODDS WITH THE "INVARIANCE" EXPLANATION

We begin by briefly exploring several common augmentations used in contrastive learning for natural images, and explore how they come into conflict with the common assumption described above. First, we observe that many common augmentations can affect the label of the input, depending on the downstream task. For example, many downstream image recognition tasks require color information (e.g. identifying bird species) or brightness (e.g. scene or time-of-day classification), implying that invariance to these characteristics would be undesirable. Yet hue shifts, greyscaling, and brightness shifts are common augmentations used in contrastive learning Chen et al. (2020b); He et al. (2020)

Second, repeated application of some augmentations causes challenges for *all* downstream tasks. For example, applying brightness shifts repeatedly results in any image turning completely black or completely white. Thus the class label cannot be truly invariant to this augmentation, since inputs from different classes can

experience an "augmentation collision" at this black or white image (this is formalized in Appendix B).[1] This argument also applies to other augmentations, including shifts in contrast[2] and random masking.

Third, some augmentations are commonly used *despite* ways of explicitly encoding invariance to them. For example, two image augmentations are *hue shifts* and *greyscaling*. Invariance to both of these augmentations can be explicitly encoded by always converting an image to greyscale. Yet doing so is not common practice because color information is still desirable for many downstream tasks.

The contradictions between the invariance rationale for augmentations in contrastive learning and these common practices suggest the need for additional explanations for the role of augmentations.

## 3 VIEWMAKER NETWORKS SUCCEED DESPITE DESTROYING LABEL INFORMATION

As another point of evidence that good views need not be label-preserving, we consider the behavior of viewmaker networks (Tamkin et al., 2021b), a generative model which produces augmentations for contrastive learning. Intuitively, viewmakers learn a stochastic augmentation policy that makes the contrastive task as hard as possible for the encoder. The stochastic augmentations are parameterized as additive perturbations bounded by an $L_1$ norm, meaning the viewmaker can alter but not completely destroy the original image.

Formally, given an input $x \in \mathbb{N}$, a viewmaker network $V_\psi$ is trained jointly with an encoder $E_\theta$ to optimize the minimax expression:

$$\max_\psi \min_\theta \mathcal{L} \left( E_\theta \left( x + \epsilon \frac{V_\psi(x, \delta_1)}{||V_\psi(x, \delta_1)||_1} \right), E_\theta \left( x + \epsilon \frac{V_\psi(x, \delta_2)}{||V_\psi(x, \delta_2)||_1} \right) \right)$$

Here $\mathcal{L}$ is a multiview loss function (e.g. (Chen et al., 2020b; He et al., 2020)), $x$ is a minibatch of inputs, $|| \cdot ||_1$ is the $L_1$ norm, $\epsilon$ is the *distortion budget* controlling the strength of the views, and $\delta_1, \delta_2 \sim N(0, 1)$ are random inputs that enable the viewmaker to learn a stochastic augmentation policy. We clamp the output of the viewmaker for images to $[0, 1]$ as in Tamkin et al. (2021b).

Viewmaker networks learn to stochastically alter different parts of the input, including task-relevant features, meaning that these augmentations are not label-preserving. Nevertheless, as we will see shortly, viewmaker networks enable strong performance on multiple downstream tasks, including often better performance than expert-designed augmentations. Moreover, this **feature dropout** capability of viewmaker networks may help them to learn many features well rather than focusing on the easiest ones.

### 3.1 DATASETS

We consider the behavior of viewmaker networks on four datasets, including three image and one audio dataset. Each dataset is constructed in such a way as to support two distinct downstream classification tasks, enabling us to examine how well each downstream task is learned. The presence of two downstream tasks enables us to analyze the *foundation model* setting where we wish to learn features relevant for multiple downstream tasks, as opposed to one set or the other.

**Image datasets** The three image datasets are based on the canonical CIFAR-10 image-recognition dataset (Krizhevsky, 2009) (MIT-License). One task is always to predict the CIFAR-10 object label (e.g. `airplane` or `bird`). The other task is dependent on an additional feature overlaid on the image: **C+Shapes:** The CIFAR-10 image is overlaid with one of three randomly-colored shapes: a square, a triangle, or a circle.

---

[1]Note that invariance is not to be confused with the related but distinct property of equivariance, often discussed as a desirable property of network architectures (e.g. see Fukushima & Miyake (1982); Chen et al. (2020a))

[2]Continuous reduction in contrast eventually produces single-color images, given finite precision images

The second task is to predict what shape was overlaid (N=3 classes). **C+Digits:** The CIFAR-10 images are overlaid with four copies of a randomly-sampled digit from the MNIST dataset. The second task is to predict the digit class (N=10 classes). **C+Letters:** The CIFAR-10 images are overlaid with four copies of a randomly-colored English letter. The second task is to predict the class of the letter (N=26 classes).

**Audio dataset**   The audio dataset is created by overlaying the audio of a spoken digit (from the AudioMNIST dataset (Becker et al., 2018), MIT License) with a random background sound (collected from one of three possible classes: cafe, machinery, and traffic) (Saki et al., 2016; Saki & Kehtarnavaz, 2016). The tasks are to predict the digit class (N=10 classes) and to predict the sound class (N=3 classes). Inputs are presented to the network as log mel spectrograms.

### 3.2   EXPERIMENTS

**Pretraining**   We pretrain with the SimCLR algorithm for 200 epochs with a batch size of 256 and a temperature of 0.1. We use a ResNet-18 model with standard modifications for smaller inputs (including a smaller stride and no initial maxpool) as used in Tamkin et al. (2021b). For the expert augmentations, we use the standard SimCLR augmentations for the image datasets (Chen et al., 2020b), and the SpecAug (Park et al., 2019) augmentations for the audio datasets, which randomly mask out different frequency and time bands, as well as the WaveAug (Kharitonov et al., 2021) augmentations, which alter various properties of the waveform such as the pitch and speed. For the viewmaker augmentations, we use a budget of $\epsilon = 0.05P$ for the image datasets, and $\epsilon = 0.125P$ for the audio datasets, where $P$ is the number of pixels in the input.

**Linear Evaluation**   We evaluate the quality of the learned representations by training a linear softmax classifier on top of the prepool representations. We train for 100 epochs, using the same parameters as Viewmaker (Tamkin et al., 2021b), training separate linear classifiers using the same pretrained network for each downstream task (Chen et al., 2020b). Augmentations are applied during training but not evaluation.

### 3.3   RESULTS

**Qualitative evidence of feature dropout**   Visually, the viewmaker augmentations seem to stochastically alter different aspects of the input, as shown in Figure 1. In addition to modifying the background of each input, the viewmaker also selectively modifies the additional synthetic features added to each domain: **C+Digits:** The viewmaker augmentations selectively add pixels to the MNIST digits, making it difficult to distinguish which number is present. **C+Shapes:** The viewmaker augmentations sometimes draw squares around the shape in the center, making it difficult to determine the shape class. **C+Letters:** The viewmaker draws letter-like markings on top of the letters, obscuring the letter identity and color. **Audio:** The viewmaker identifies the narrow band corresponding to the speech and applies perturbations to it. As can be seen in Figure 1, these label-destroying augmentations are quite common, occuring in a sizeable fraction of the sampled views.

**Quantitative evidence of feature dropout**   We also measure this selectivity of features quantitatively in Appendix C.2 and Figure 4. We augment images 1,200 times and observe the impact on the predictive probability of the correct object class. Two clear modes appear for the viewmaker augmentations, but not expert augmentations. This corresponds to the fraction of time the viewmaker destroys the overlaid feature information (low P(correct object class)) and preserves it (high P(correct object class)).

**Viewmaker succeeds despite destroying label information**   As shown in Table 1 and Table 2, viewmaker networks are able to achieve good accuracy on both tasks, while expert augmentations frequently achieve lower performance on one or both tasks. For example, on the image tasks, while expert views achieve slightly

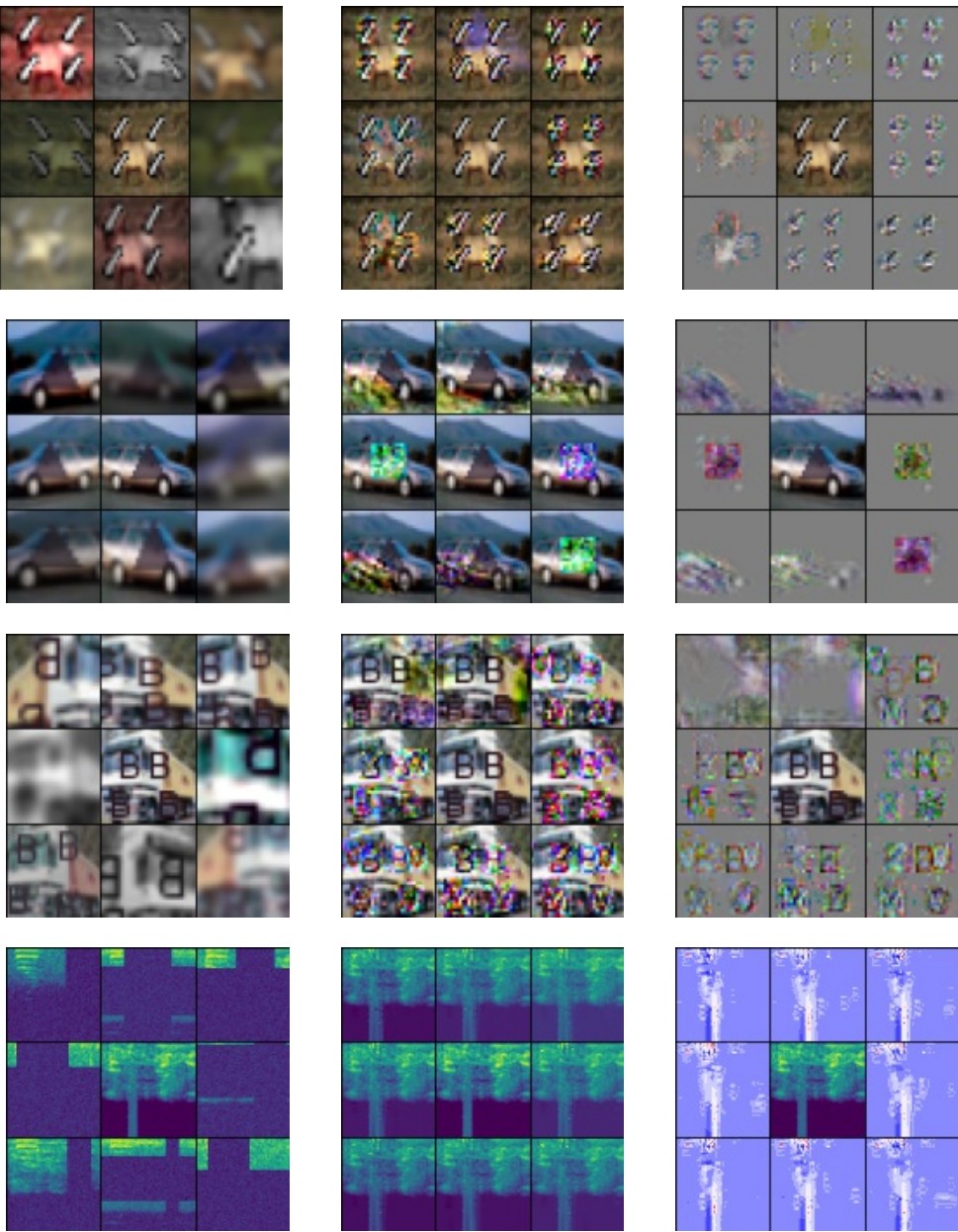

Figure 1: **Comparison of viewmaker and expert augmentations on datasets with multiple features.** The viewmaker augmentations adapt to the particular semantics of the input data, and make targeted perturbations which remove the class-relevant information of the synthetic features (e.g. occluding the digit, shape, letter, or speech). Despite this, the encoder network is still able to learn strong representations. *Rows* (from top): Digits, Shapes, Letters, Audio. *Columns* (from left): Expert augmentations, viewmaker augmentations, difference between original and viewmaker augmentation, rescaled to [0,1]. Center image in each grid is the original. Audio Expert views shown are Spectral views.

|  | Viewmaker (CIFAR-10) | Expert (CIFAR-10) | Viewmaker (Object) | Expert (Object) |
|---|---|---|---|---|
| CIFAR-10 Only | 84.5 | **86.2** | - | - |
| C+Shape | **79.8** | 76.0 | **100.0** | **100.0** |
| C+Digit | **69.3** | 58.8 | **94.3** | 86.7 |
| C+Letter | 71.9 | **74.8** | **96.9** | 94.1 |

Table 1: **Transfer accuracy on different features.** Viewmaker networks are able to achieve good performance across multiple downstream tasks, while expert views sometimes falter. Networks are pretrained on the datasets on the left, and transfer accuracy is reported for the different conditions on the columns. Runs are averages of three seeds (with the exception of CIFAR-10 Only, which is taken from (Tamkin et al., 2021b)).

|  | Speech Accuracy | | | Background Sound Accuracy | | |
|---|---|---|---|---|---|---|
|  | Viewmaker | Spectral | Waveform | Viewmaker | Spectral | Waveform |
| Speech Only | 92.4 | **97.0** | 76.7 | - | - | - |
| Bkgd. Sound Only | - | - | - | **100.0** | 32.64 | **100.0** |
| Speech + Sound | **60.8** | 10.1 | 53.6 | **97.0** | 47.2 | 43.3 |

Table 2: **Audio transfer accuracies.** Viewmaker networks achieve good performance across multiple tasks, while expert views sometimes suffer catastrophic drops as another feature is added. Networks are pretrained on the datasets on the left, and transfer accuracy is reported for the different conditions on the columns. Runs are averages of three seeds.

higher performance on the image only, they experience a large drop in accuracy when the synthetic feature is added. In two of these cases (Shape and Digit) the viewmaker models are able to achieve a higher accuracy on both the image and the synthetic feature, while on the third (Letters) viewmakers achieve slightly lower accuracy on the images but achieve half the error on the synthetic object. For the audio experiments the picture is similar—the viewmaker is able to avoid catastrophic drops in performance learning both features together, achieving the highest accuracy on both, while the expert views experience larger drops and worse overall performance. Note that the high performance of expert views for our control tasks (CIFAR-10/Speech/Sound Only) indicates that the viewmaker views are not merely better all-around views, but that they specifically help the model learn multiple features.

These results provide additional evidence that label-preserving views are not necessary for learning good representations—and that the ability to perform feature dropout may improve learning of multiple tasks.

## 4 THEORETICAL ANALYSIS OF FEATURE INTERACTIONS IN A LINEAR CONTRASTIVE SETTING

In this section, we theoretically analyze a simple linear model that captures the essence of how label-destroying augmentations can improve downstream accuracy. We study a setting where the data contains many underlying features that are relevant to downstream classification tasks, and where these features are preserved to varying degrees across augmentations. We will show that a linear model trained with a contrastive objective learns these features, and that adding noise to one feature can speed the learning of other features during gradient descent. One difference between the linear setting we theoretically analyze and Section 3 is that in this section we add stochastic Gaussian noise to destroy features across augmentations, as opposed to the more bimodal feature dropout behavior seen in Figure 1.

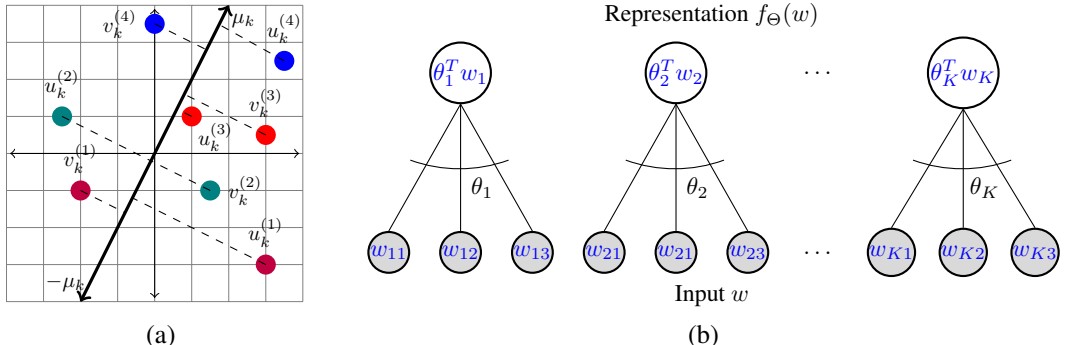

(a)                                             (b)

Figure 2: We show how label-destroying augmentations can aid learning of other features in a linear contrastive setting: (a) The correlation of the $k$th feature of an augmentation pair, shown for $d = 2$. Each pair $u_k^{(i)}$ and $v_k^{(i)}$ have correlated projections onto the ground truth $\mu_k$ direction, representing the feature conserved across augmentations. (b) Feedforward linear network which computes the representation $f_\Theta(w)$. As each feature $\mu_k$ is learned ($\theta_k \to \mu_k$) the representations of the two views $f_\Theta(u^{(i)}), f_\Theta(v^{(i)})$ become more similar, decreasing the contrastive loss.

### 4.1 DATA MODEL AND SETTING

We study a model which consists of data with $K$ distinct features, each corresponding to some ground truth unit-vector directions $\mu_1, \ldots, \mu_K \in \mathbb{R}^d$. We sample each data point $u \in \mathbb{R}^{K \times d}$ and its *augmentation* (a.k.a. its *positive pair* or its *view*) $v \in \mathbb{R}^{K \times d}$ as follows. For $k \in 1, \ldots, K$, the $k$th row of $u$, which we denote $u_k$, is drawn from the Gaussian distribution $\mathcal{N}(0, I_d)$. The $k$th row of the augmentation, $v_k$, is drawn from the same distribution, but is correlated with $u_k$ in the $\mu_k$-direction (and is otherwise independent in the other directions). The strength of the correlation is governed by parameter $\alpha_k \in [0, 1]$ in the following sense: $v_k^T \mu_k = \alpha_k u_k^T \mu_k + \sqrt{1 - \alpha_k^2} \xi$, where $\xi \sim \mathcal{N}(0, 1)$. Thus the larger $\alpha_k$, the stronger the correlation in that feature across the two views. Figure 2(a) visualizes the correlation of $(u_k, v_k)$ in an augmented pair. Formally, we can write that $(u_k, v_k) \sim \mathcal{N}\left(0, \begin{pmatrix} I_d & \alpha_k \mu_k \mu_k^T \\ \alpha_k \mu_k \mu_k^T & I_d \end{pmatrix}\right)$, for a vector $\boldsymbol{\alpha} \in [0, 1]^k$.

We will learn a model $\Theta \in \mathbb{R}^{K \times d}$, which represents a collection of $K$ feature extractors, as pictured in Figure 2(b). The model $\Theta$, with rows $\{\theta_k\}_{k \in [K]}$, maps a data point $w \in \mathbb{R}^{K \times d}$ to a representation $f_\Theta(w) \in \mathbb{R}^K$ by computing a score $w_k^T \theta_k$ for each element in the representation. That is, $(f_\Theta(w))_k = w_k^T \theta_k$.

Our goal is that the model $\Theta$ will be useful for a downstream classification task which depends on the ground truth features. A good representation will capture ground truth features that are correlated across augmentations, such that $\theta_k$ is aligned with $\mu_k$ or $-\mu_k$.

**Training.** We will study the the evolution of $\Theta$ as we optimize a standard constrictive learning objective using gradient descent (Dosovitskiy et al., 2014; Chen et al., 2020b). At each round of gradient descent, we sample a fresh batch of $m$ data points and their augmentations, $(U, V) := \{(u^{(i)}, v^{(i)})\}_{i \in [m]}$. For each $i, j \in [m]$, we compute a similarity score $z_{ij} := \langle f_\Theta(u^{(i)}), f_\Theta(v^{(j)}) \rangle = \sum_k (\theta_k^T u_k^{(i)})(\theta_k^T v_k^{(j)})$ using the dot product of their $K$-dimensional representations. We then compute the logits $p_{ij} := \frac{\exp(z_{ij})}{\sum_{j'} \exp(z_{ij'})}$ using the softmax function, and use the classwise cross entropy loss function $\mathcal{L}(\Theta; U, V) := -\log(p_{ii})$.

## 4.2 MAIN RESULT

We will study gradient descent (GD) on the cross entropy loss, and consider how adding noise to one feature affects the learning of the other features. As suggested earlier, we can measure how well we learn the $k$th feature by measuring the alignment of $\theta_k$ with $\mu_k$ or $-\mu_k$. A natural way to measure this alignment is the acute angle between $\pm\mu_k$ and $\theta_k$, given by $\arccos\left(\frac{|\mu_k^T\theta_k|}{\|\theta_k\|_2}\right)$. Lemma E.1 in Appendix E proves that this quantity directly determines the test accuracy on a natural downstream linear classification task.

Formally, we say we *add noise* to some feature $k'$ of a data point $v$, if for some $\beta \in [0,1)$, we let $\tilde{v}_{k'} = \beta v_{k'} + \sqrt{1-\beta^2}\xi$, where $\xi \sim \mathcal{N}(0, I_d)$, and $\tilde{v}_k = v_k$ for $k \neq k'$. Thus if $(u,v)$ were a pair generated with the correlation coefficients $\{\alpha_k\}_{k\in[K]}$, then the distribution of $(u,\tilde{v})$ comes from the modified correlation coefficients $\{\tilde{\alpha}\}_{k\in[K]}$ with the single modification $\tilde{\alpha}_{k'} = \beta\alpha_k$. We now present our main theorem:

**Theorem 4.1** (Noise improves feature learning). *There exists a universal constant $C$, such that the following holds. Let $\Theta^{(t+1)} = \Theta^{(t)} - \eta(\nabla\mathcal{L}(U,V;\Theta) + \lambda\Theta^{(t)})$, and $\tilde{\Theta}^{(t+1)} = \Theta^{(t)} - \eta(\nabla\mathcal{L}(U,\tilde{V};\Theta) + \lambda\Theta^{(t)})$, where $\tilde{V}$ is $V$ with any amount of added noise in the $k'$ feature. This has the effect of changing $\alpha_{k'}$ to $\tilde{\alpha}_{k'}$ for any $\tilde{\alpha}_{k'} < \alpha_{k'}$. Then for any $k \neq k'$, if $|\theta_k^T\mu_k| \leq \frac{1-\alpha_{k'}^2}{C}\|\theta_k\|$, $\|\theta_{k'}\|^3 \leq |\theta_{k'}^T\mu_k|$, and $\|\theta_k\|^2 \leq \frac{\alpha_k(1-\alpha_{k'}^2)}{C}$, then for a small enough step size $\eta$, $\mathbb{E}_{U,V}\left[\arccos\left(\frac{|\mu_k^T\theta_k^{(t+1)}|}{\|\theta_k^{(t+1)}\|_2}\right)\right] > \mathbb{E}_{U,\tilde{V}}\left[\arccos\left(\frac{|\mu_k^T\tilde{\theta}_k^{(t+1)}|}{\|\tilde{\theta}_k^{(t+1)}\|_2}\right)\right].$*

We briefly comment on the three assumptions on $\Theta$ in the theorem. The first assumption, $|\theta_k^T\mu_k| \leq \frac{1-\alpha_{k'}^2}{C}\|\theta_k\|$ requires that $\theta_k$ is not too aligned with $\mu_k$ – that is, the result applies to all features $k$ that aren't already learned too well. The second two assumptions are satisfied if the $k'$th feature has been learned to some extent, and the norm of $\theta_k$ and $\theta_{k'}$ are small, which can be enforced throughout training with $\ell_2$ regularization.

The theorem guarantees that at any point in training, if we add noise to the $k'$th feature, the next step of GD learns all other features *better* than if we didn't add noise. To validate the implication of this result for the complete trajectory of GD, we include simulations in Appendix D. Our experiments show that introducing noise part-way through training to dominant features can significantly speed the alignment of weak features, with only a small cost to the alignment of the dominant features. We prove our result in Appendix E, including intuition and a technical overview of the steps in Section E.3.

## 5 RELATED WORK

**Understanding contrastive and multiview learning**    Many prior works have laid the foundations for current contrastive and multiview learning algorithms (Becker & Hinton, 1992; Hadsell et al., 2006; Dosovitskiy et al., 2014; Wu et al., 2018; Bachman et al., 2019; Misra & van der Maaten, 2020; He et al., 2020; Chen et al., 2020b). Several works perform analysis studies of contrastive learning to identify important factors (Cole et al., 2021; Zhao et al., 2021) or how contrastive models differ from supervised learning (Yang et al., 2020; Ericsson et al., 2021a; Karthik et al., 2021). HaoChen et al. (2021) study contrastive learning using the concept of an augmentation graph. This model assumes the fraction of non-label preserving augmentations is "extremely small;" interestingly, we show in practice it can quite large and still yield good performance. Wang et al. (2022) theoretically study contrastive learning under an assumption of label-preserving augmentations, though they show that such an assumption alone does not suffice to learn. Most relevant to our work, Tian et al. (2020); Ericsson et al. (2021b) study how the information shared between different views impacts learning of downstream tasks. We complicate this picture by analyzing the foundation model setting where a single model must learn features that are useful for multiple tasks that are not known in advance. In this setting, we find that label-destroying perturbations, thought to be harmful by Tian et al. (2020), are useful for preventing one feature from suppressing others.

**Feature suppression**  Our work is closely connected to the notion of *feature suppression* (Hermann & Lampinen, 2020), where the presence of one feature can crowd out or suppress the learning of other features. Several works have explored the relevance of this concept in contrastive learnings. For example, the original SimCLR paper (Chen et al., 2020b) noted that color jitter augmentation was necessary to prevent the network for using only the color profile of the input to solve the contrastive task. Followup work (Chen et al., 2021) explores this phenomenon in more detail, characterizing how different hyperparameters and dataset features affect feature suppression. Other works have attempted to address feature suppression in contrastive learning, either via auxiliary losses (Li et al., 2020) or by modifying representations in the latent space (Robinson et al., 2021). Our work relates to these in two ways. First, we empirically and theoretically investigate feature suppression as an alternate rationale for the role of augmentations, as opposed to invariance. Second, we show that an existing method, viewmaker networks (Tamkin et al., 2021b), can identify and potentially neutralize suppressing features in an interpretable way, resulting in better performance than expert augmentations.

**Spurious correlations and shortcut features**  Outside the framing of feature suppression, several other works explore how classifiers can learn or make use of unwanted features. Shortcut features (Geirhos et al., 2020) describe often-simple features (e.g. the average color of an input) which are learned by networks at the expense of more salient features (e.g. the object class). This notion is connected to spurious correlations (Simon, 1954) in deep learning which have been explored extensively (Sagawa et al., 2019; 2020; Srivastava et al., 2020; Tu et al., 2020; Xiao et al., 2021), including in the context of self-supervised learning (Minderer et al., 2020). Other works have also performed theoretical analysis of how related dynamics affect learning in the supervised setting (Li et al., 2019; Shah et al., 2020). Our work suggests that viewmaker networks may be a useful tool as well here—both as an interpretability tool to visualize the different features a network relies on, and as a way to reduce reliance on particular features without completely destroying the information.

## 6  DISCUSSION AND CONCLUSION

We have presented several different arguments against the commonly-articulated belief that the main role of augmentations is to specify invariances for a contrastive learning model. First, common augmentations such as brightness shifts would result in useless representations if networks became truly invariant to them. Second, viewmaker networks succeed at contrastive learning despite learning label-destroying perturbations which drop out different features in the input. Finally, we present an analysis of a linear contrastive setting where we prove that label-destroying views actually have a positive effect on contrastive learning if the goal is to avoid learning one feature at the expense of others.

Our work has limitations. For example, our empirical analysis is limited to four synthetic datasets spanning vision and audio, whereas self-supervised learning may be applied to naturalistic data spanning a much wider range of modalities (Tamkin et al., 2021a; 2022). In addition, our theoretical analysis considers a linear contrastive setting, whereas current neural networks are highly nonlinear. Improving upon both of these fronts is an exciting area for future work.

On the other hand, understanding augmentations as dropping out easy features suggests possible ways of improving the performance of self-supervised learning. For instance, viewmaker networks cap the extent to which views can differ from the underlying image. Our analysis here suggests the role of this cap indirectly sets the dropout rate of different features in the input; some way of directly encoding this objective may yield more flexible and performant viewmaker approaches.

The challenge of learning a broad range of useful features lies at heart of self-supervised learning. We hope our work sheds light on this challenge in contrastive learning, especially as these objectives continue to develop and are applied more broadly and at larger scale.

**Ethics Statement**   Our work is centered on conceptual understanding, making it challenging to confidently predict societal impacts. Better conceptual understanding of existing methods may help us understand the failure modes and successes of current models better, which may have positive impacts. However, if this understanding enables the development of more powerful methods, the work may indirectly accentuate whatever social impacts (positive or negative) those applications have.

**Reproducibility Statement**   We include hyperparameters and experimental settings for our experiments in Section 3, complete statements of our theoretical results in Appendix E, and will release our codebase to reproduce our results.

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

## A    CODE RELEASE

We are preparing our code and will release it on a public GitHub repo.

## B    FORMALIZATION OF OBSERVATION IN SECTION 2

**Definition B.1** (Invariance). *A function $f : \mathbb{R}^m \to \mathbb{R}^n$ is invariant to a set of transformations $G$ if and only if $f \circ g(x) = f(x)$ for all $x \in \mathbb{R}^m$ and for all $g \in G$.*

**Definition B.2** (Augmentation collision). *An augmentation collision occurs if, for two inputs $x_a, x_b$ and set of transformations $G$, there exist $g_a^{(1)}, \ldots, g_a^{(n_a)}, g_b^{(1)}, \ldots, g_b^{(n_b)} \in G$ for some $n_a, n_b \in \mathbb{N}$ such that $g_a^{(1)} \circ \ldots \circ g_a^{(n_a)}(x_a) = g_b^{(1)} \circ \ldots \circ g_a^{(n_b)}(x_b)$.*

**Observation B.3.** *If there exists an augmentation collision for inputs $x_a, x_b$ and transformation set $G$, and $f$ is invariant to $G$, then $f(x_a) = f(x_b)$.*

*Proof.* By the definition of an augmentation collision, $g_a^{(1)} \circ \ldots \circ g_a^{(n_a)}(x_a) = g_b^{(1)} \circ \ldots \circ g_a^{(n_b)}(x_b)$. By the definition of a function, we have $f \circ g_a^{(1)} \circ \ldots \circ g_a^{(n_a)}(x_a) = f \circ g_b^{(1)} \circ \ldots \circ g_a^{(n_b)}(x_b)$. Applying invariance, we obtain $f(x_a) = f(x_b)$. □

Applying this observation, we observe that if the downstream labeling function $f$ is invariant to a class of augmentations, then there cannot be an augmentation collision for inputs with different labels. However, common augmentations such as brightness shifts can reduce any image to a black or white image, resulting in an augmentation collapse between any two inputs.

## C    ADDITIONAL FEATURE DROPOUT EXPERIMENTS

### C.1    QUANTIFYING THE IMPORTANCE OF FEATURE DROPOUT

To assess the importance of label-destroying augmentations to the success of the viewmaker, we experiment with a setup where the viewmaker cannot destroy the information in the object class. To do this, we compute a mask around the object and zero out any perturbation from the viewmaker within that mask. We then perform pretraining and transfer as usual.

As we report in Table 3, the accuracy of the CIFAR-10 class label drops precipitously, as expected. At the same time, the accuracy of two of the other objects remains mostly constant (shape and digits), while the accuracy for letters declines modestly (perhaps because the color of the letter is now able to suppress the learning of the letter class.

| | Viewmaker (C-10) | Mask-Viewmaker (C-10) | Viewmaker (Object) | Mask-Viewmaker (Object) |
|---|---|---|---|---|
| C+Shape | 79.8 | 26.0 | 100.0 | 95.8 |
| C+Digit | 69.3 | 50.7 | 94.3 | 95.0 |
| C+Letter | 71.9 | 23.2 | 96.9 | 71.8 |

Table 3: **Experiments with a masked viewmaker which is unable to destroy the object class.** Transfer accuracy on CIFAR-10 (C-10) and the object task (Shape, Digit, or Letter). The Mask-Viewmaker has its perturbation masked such that it cannot destroy the label of the object. This results in the features in the object suppressing the CIFAR-10 accuracy, while leaving the object accuracy relatively unscathed.

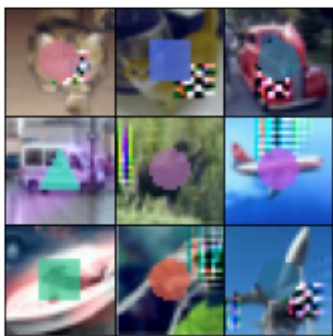 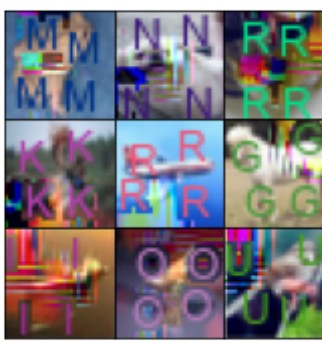 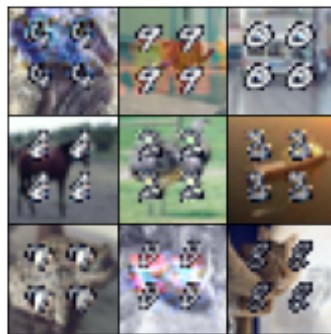

Figure 3: **Non-label destroying Viewmaker perturbation examples.**

### C.2 QUANTIFYING THE DEGREE OF FEATURE DROPOUT

We perform an exploratory analysis to testing how well different views drop out the features in an input. We augment a 1,200 examples (CIFAR-10 image plus an overlaid object) using a given augmentation policy (either the expert or viewmaker augmentations). We then encode the model with a classifier trained off of the other augmentation policy (i.e. expert for viewmaker augmentations or the reverse) in order to test how well the augmentations drop out the features. We use a different encoder to see the effects of the augmentations prior to the encoder having a chance to adapt to them.

We observe a bimodal behavior for the viewmaker views, shown in Figure 4, suggesting that the model is adapting to the semantics of the input and has learned to stochastically drop out the simple feature some fraction of the time. By contrast, the expert views display no such structure. Using the corresponding encoder and views leads to models performing uniformly well, as shown in Figure 5.

### D END-TO-END SIMULATIONS OF LINEAR SETTING

We empirically test the performance of the full trajectory of gradient descent when we add noise to the data. We study a setting with one weak feature with correlations coefficient $\alpha_1 \leq 0.5$, and 50 dominant features with $\alpha_k = 1$ for $k = 2, \cdots, 51$. We compare two approaches run on the same data: in the first approach, we run 150 iterations of GD without adding noise. In the second, we run 50 iterations of GD without noise, and then add noise to the dominant features for the remaining 100 iterations.

In Figure 6(top), we compare the alignment of Feature 1 (the weak feature) and Feature 2 (one of the dominant features) to the ground truth in the two approaches. We observe that adding noise consistently accelerates the

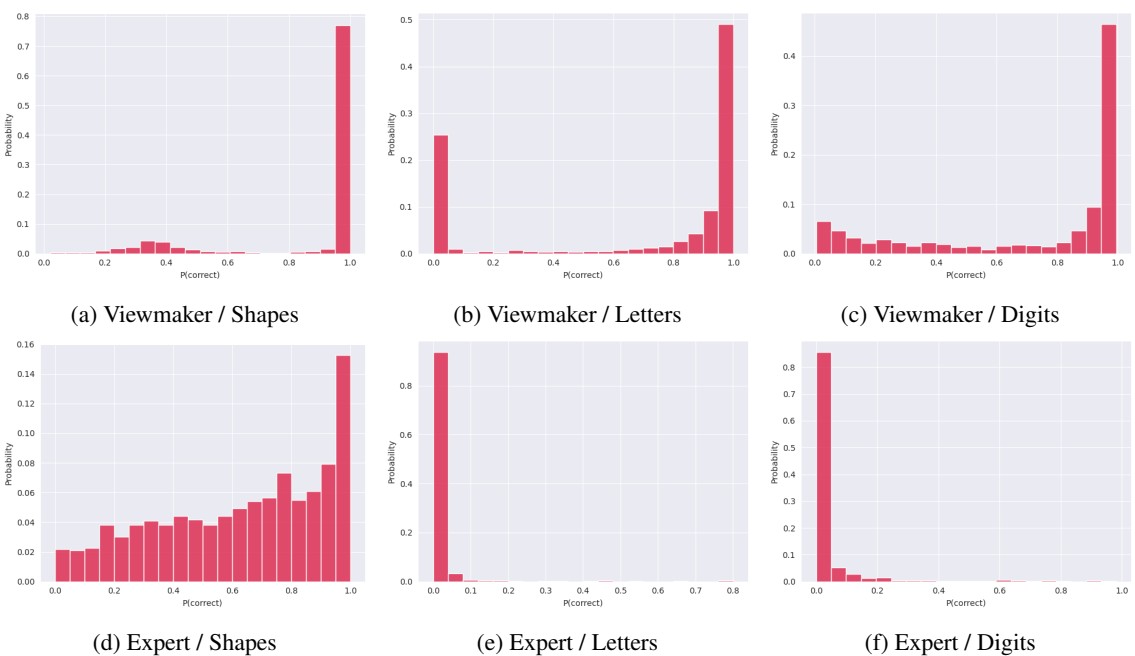

(a) Viewmaker / Shapes      (b) Viewmaker / Letters      (c) Viewmaker / Digits

(d) Expert / Shapes      (e) Expert / Letters      (f) Expert / Digits

Figure 4: **Viewmaker augmentations stochastically drop out simple features added to the input.** Probability of the correct answer for different augmentations (Viewmaker or Expert) and different examples from different datasets (Shapes, Letters, Digits). Each histogram shows a single example from each dataset randomly augmented 1200 times, and the corresponding probabilities of the correct answer. The viewmaker augmentations display a bimodal structure, indicating that the simple feature is selectively either destroyed or preserved. The expert augmentations by contrast lack such structure, reflecting their lack of adaptation to the structure of each input.

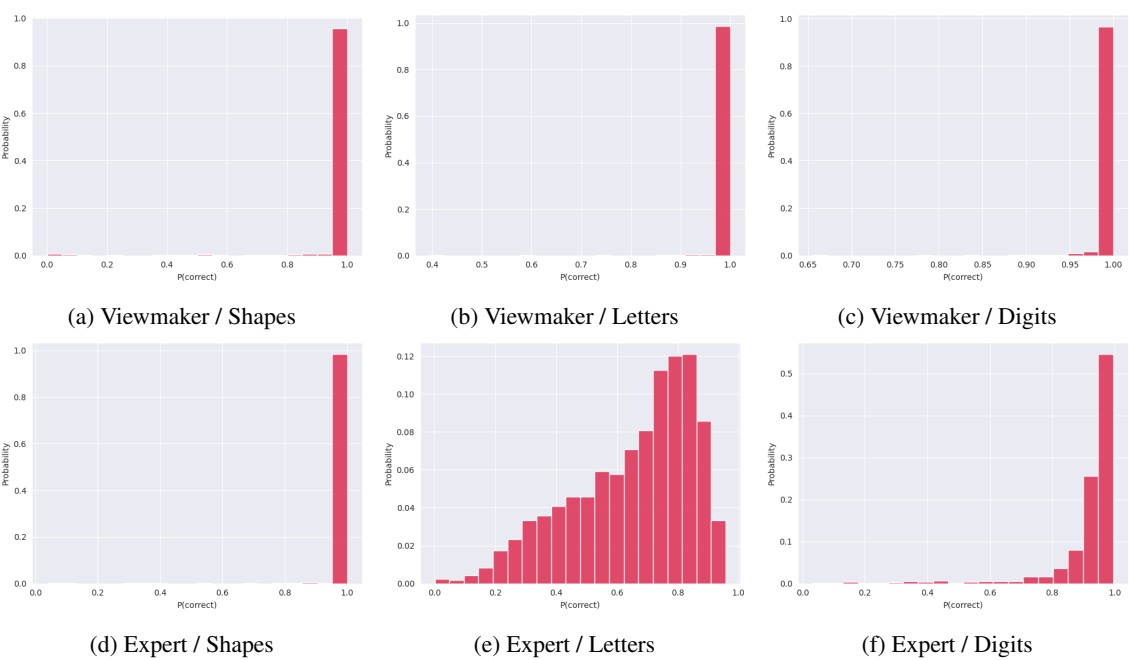

(a) Viewmaker / Shapes     (b) Viewmaker / Letters     (c) Viewmaker / Digits

(d) Expert / Shapes     (e) Expert / Letters     (f) Expert / Digits

Figure 5: **Evaluating views with their respective encoder does not reveal bimodal structure for view-maker or expert views.** Details are the same as in Figure 4, with the exception that views are evaluated on their corresponding encoder.

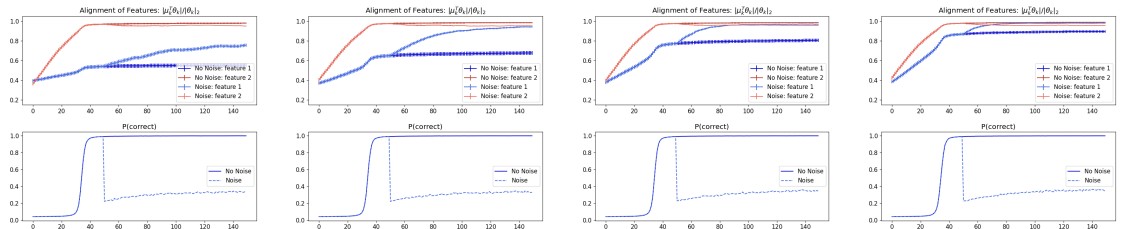

Figure 6: Alignment of features with verses without added noise. From left to right: $\alpha_1 = 0.125, 0.25, 0.375, 0.5$. The top plots show the alignment of Features 1 (weak) and 2 (dominant) to the ground truth; the bottom plots shows the probability of predicting the correct augmentation pair from the batch. Standard deviation bars are shown for the mean alignment over 200 runs. We used dimension $d = 5$, and a batch size of $m = 25$.

learning of the weak feature (blue), with little cost to the dominant features (red). The affect is consistent among many choices for $\alpha_1$, the correlation coefficient of the weak feature. We also plot in Figure 6(bottom) the probability of predicting the correct class (pair) of the view under both approaches. We observe that this probability drops sharply when we add noise, which we believe is the mechanism for faster learning with noise.

We remark that we chose to add noise to all the dominant features (instead of a single $k'$ a in our theorem) to accentuate the effect of adding noise. We observed a similar effect, but smaller, when we added noise to fewer features, or when there were fewer than 50 dominant features.

## E    FULL PROOFS OF PROPOSITIONS AND THEOREMS

We begin by stating and proving Lemma E.1 on the downstream classification accuracy.

**Lemma E.1** (Downstream classification accuracy). *Suppose we draw labeled data points* $(u, y) \in \mathbb{R}^{K \times d} \times \{+1, 1\}$*, where as before,* $u_k \sim \mathcal{N}(0, I_d)$ *for* $k \in [K]$*, and the label is given by* $\mathrm{sign}(u_k^T \mu_k)$*. Then the best linear classifier* $\boldsymbol{a} \in \mathbb{R}^K$ *on the representations* $f_\Theta(u) \in \mathbb{R}^K$ *achieves an test error of* $\frac{1}{\pi} \arccos\left(\frac{|\mu_k^T \theta_k|}{\|\theta_k\|_2}\right)$*. That is*

$$\min_{\boldsymbol{a} \in \mathbb{R}^K} \Pr_u[\mathrm{sign}(\boldsymbol{a}^T f_\Theta(u)) \neq \mathrm{sign}(\mu_k^T u_k)] = \frac{\arccos\left(\frac{|\mu_k^T \theta_k|}{\|\theta_k\|_2}\right)}{\pi}. \tag{1}$$

*Thus if* $\theta_k$ *and* $\mu_k$ *are orthogonal, then the test error is* $50\%$*. If the angle between* $\theta_k$ *and the* $\pm\mu_k$ *is zero, then we achieve perfect classification accuracy.*

*Proof.* It is easy to see that the best linear classifier $\boldsymbol{a}$ will (up to scaling) be equal to the vector $\mathrm{sign}(\mu_k^T \theta_k) e_k$. Such a classifier predicts the correct sign whenever $\mathrm{sign}(\boldsymbol{a}^T f_\Theta(u)) = \mathrm{sign}(\mu_k^T \theta_k) \mathrm{sign}(\theta_k^T u_k)$ equals $\mathrm{sign}(\mu_k^T u_k)$, which occurs exactly a $1 - \frac{\arccos\left(\frac{|\mu_k^T \theta_k|}{\|\theta_k\|_2}\right)}{\pi}$ fraction of the time.  $\square$

In the rest of this section, we prove our main theoretical result, Theorem 4.1, which shows that $\arccos\left(\frac{|\mu_k^T \theta_k|}{\|\theta_k\|_2}\right)$ decreases faster in expectation during gradient descent if we add noise to the $k'$ feature.

### E.1 NOTATION.

We let $\delta_{ij}$ denote the $\delta$-function which equals 1 if $i = j$ and 0 otherwise. For a parameter $\Theta = \{\theta_k\}_{k \in [K]}$, we let $\theta_k^{\|} := \mu_k \mu_k^T \theta_k$ be the projection of $\theta_k$ in the $\mu_k$ direction. We let $\theta_k^{\perp} = \theta_k - \theta_k^{\|}$ be the projection of $\theta_k$ orthogonal to the feature $\mu_k$.

Throughout this section, we consider the ground truth directions to be fixed, and we fix some initial correlation vector $\boldsymbol{\alpha}$. We let $\mathbb{P}_{\boldsymbol{\alpha}}$ denote the distribution from which the pair $(u, v)$ is drawn from the Gaussian distribution described in Section 4 with correlation coefficients $\boldsymbol{\alpha}$. When unspecified, the variables $U, V$ are drawn from the distribution $\mathbb{P}_{\boldsymbol{\alpha}}^m$. Since we study what happens when we vary $\alpha_{k'}$, for $x \in [0, 1]$, we use the shorthand $\mathbb{P}_x$ to denote the distribution $\mathbb{P}_{\boldsymbol{\alpha}(x)}^m$, where $\alpha(x)_{k'} = x$, and $\alpha(x)_k = \alpha_k$ for all other $k$.

We denote $\mathcal{L}_i(\Theta; U, V) = \mathrm{CE}(\{p_{ij}\}_{j \in [m]}, e_i) = -\log(p_{ii})$, which we abbreviate by $\mathcal{L}_i$. When it is clear that we are considering $\mathcal{L}_i$ for some fixed $i$, we omit the superscripts on the $i$th data point or its pair. That is, we denote $u_k := u_k^{(i)}$ and $v_k := v_k^{(i)}$.

### E.2 PRELIMINARIES

The following facts about of the derivative of the cross entropy loss are easy derived.

**Lemma E.2.**
$$\frac{\partial \mathcal{L}_i}{\partial \Theta} = \sum_j (p_{ij} - \delta_{ij}) \frac{\partial z_{ij}}{\partial \Theta} = \sum_i \sum_{j \neq i} p_{ij} \left( \frac{\partial z_{ij}}{\partial \Theta} - \frac{\partial z_{ii}}{\partial \Theta} \right), \tag{2}$$

*where*
$$\frac{\partial z_{ij}}{\partial \theta_k} = (u_k^{(i)} v_k^{(j)T} + v_k^{(j)} u_k^{(i)T}) \theta_k. \tag{3}$$

We will also need the following facts on Gaussian random variables. The first, Stein's Lemma, is well known.

**Lemma E.3** (Stein's Lemma).
$$\mathbb{E}_{X \sim \mathcal{N}(0,\sigma^2)}[X f(X)] = \sigma^2 \mathbb{E}_{X \sim \mathcal{N}(0,\sigma^2)}[f'(X)]. \tag{4}$$

The next two lemmas are proved in Section E.4.

**Lemma E.4.** *There exists some constant $C$ such that following holds. If $\sigma \leq \frac{1}{C}$, and $0 \leq t \leq \frac{1}{\sigma}$, then for any $c \in \{0, 1, 2, 3\}$, and $X \sim \mathcal{N}(0, \sigma^2)$ we have*
$$\mathbb{E}_X \left[ |X|^c \exp(t|X|) \exp(tX^2) \right] \leq C\sigma^c. \tag{5}$$

*If additionally $d \in \{0, 1, 2, 3\}$, $\rho \leq \frac{1}{C}$ and $Y \sim \mathcal{N}(0, \rho^2)$, then*
$$\mathbb{E}_X \left[ |X|^c |Y|^d \exp(t|X|) \exp(|XY|) \right] \leq C\sigma^c \rho^d. \tag{6}$$

**Lemma E.5.** *For some universal constant $C$, for any $\sigma \in [0, 1]$, $t \geq 0$, $c \in \{0, 1, 2, 3, 4\}$, we have*
$$\mathbb{E}_{X \sim \mathcal{N}(0,\sigma^2)} \left[ (\exp(t|X|) - 1) |X|^c \right] \leq Ct\sigma^c.$$

### E.3 APPROACH AND LEMMAS

**Intuition for proof of Theorem 4.1.** Our proof involves comparing the gradient of the loss in the $\theta_k$ direction, $\nabla_k := \frac{\partial}{\partial \theta_k} \mathcal{L}$ in the setting with noise to the setting without noise. Loosely, our goal is to show that for any $k$, the projection of the this gradient onto the ground truth direction, $\mu_k^T \nabla_k \mathrm{sign}(\mu_k^T \theta_k)$, increases

when when increase the noise. The main intuition comes from an expansion of this gradient in Lemma E.7, which shows that $\mathbb{E}\mu_k^T \nabla_k \operatorname{sign}(\mu_k^T \theta_k)$ approximately scales with $\sum_i (1 - p_{ii})$. Now observe that $p_{ii}$, the probability of correctly matching the $i$th view to its pair, decreases when we add noise to feature $k'$. Thus adding noise will increase $\mu_k^T \nabla_k \operatorname{sign}(\mu_k^T \theta_k)$, thereby improving the alignment.

In the remainder of this section, we outline our proof of Theorem 4.1 in this section. We prove all the lemmas below in Section E.4.

To understand $\mathbb{E}_{U,V}\left[\arccos\left(\frac{|\mu_k^T \theta_k^{(t+1)}|}{\|\theta_k^{(t+1)}\|_2}\right)\right]$ for a small enough step size, we first claim that it suffices to understand the expected projection of the gradient with respect to $\theta_k$ in the $\mu_k$ direction and in the $\theta_k$ direction. We use the notation $\nabla_k = \frac{\partial \mathcal{L}(\Theta; U, V)}{\partial \theta_k}$.

**Lemma E.6.** *Let $\theta_k^+ = \theta_k - \eta(\nabla_k + \lambda \theta_k)$. Then*

$$\lim_{\eta \to 0} \frac{1}{\eta}\left(\mathbb{E}_{U,V}\left[\arccos\left(\frac{|\mu_k^T \theta_k^+|}{\|\theta_k^+\|_2}\right)\right] - \arccos\left(\frac{|\mu_k^T \theta_k|}{\|\theta_k\|_2}\right)\right) = N\mathbb{E}_{U,V}\left[-(\mu_k^T \theta_k)(\mu_k^T \nabla_k) + \frac{\theta_k^T \nabla_k (\mu_k^T \theta_k)^2}{\|\theta_k\|_2^2}\right],$$
(7)

*where $N$ is some negative value that depends only on $\theta_k$.*

Now, since we care about the quantity $\mathbb{E}_{U,V}\left[\arccos\left(\frac{|\mu_k^T \theta_k^{(t+1)}|}{\|\theta_k^{(t+1)}\|_2}\right)\right] - \mathbb{E}_{U,\tilde{V}}\left[\arccos\left(\frac{|\mu_k^T \tilde{\theta}_k^{(t+1)}|}{\|\tilde{\theta}_k^{(t+1)}\|_2}\right)\right]$ being positive, it suffices to show that *derivative*

$$\frac{d}{dx}\mathbb{E}_{U,V \sim \mathbb{P}_x}\left[-(\mu_k^T \theta_k)(\mu_k^T \nabla_k) + \frac{\theta_k^T \nabla_k (\mu_k^T \theta_k)^2}{\|\theta_k\|_2^2}\right],$$

is negative for all $x \in [\tilde{\alpha}_{k'}, \alpha_{k'}]$. Indeed, from Lemma E.6, we have that

$$\lim_{\eta \to 0} \frac{1}{\eta}\left(\mathbb{E}_{U,V \sim \mathbb{P}_{\alpha_{k'}}}\left[\arccos\left(\frac{|\mu_k^T \theta_k^+|}{\|\theta_k^+\|_2}\right)\right] - \mathbb{E}_{U,V \sim \mathbb{P}_{\tilde{\alpha}_{k'}}}\left[\arccos\left(\frac{|\mu_k^T \theta_k|}{\|\theta_k\|_2}\right)\right]\right)$$
(8)

$$= N\int_{\tilde{\alpha}_{k'}}^{\alpha_{k'}} \frac{d}{dx}\mathbb{E}_{U,V \sim \mathbb{P}_x}\left[-(\mu_k^T \theta_k)(\mu_k^T \nabla_k) + \frac{\theta_k^T \nabla_k (\mu_k^T \theta_k)^2}{\|\theta_k\|_2^2}\right] dx,$$
(9)

so if the derivative is negative for the full range, then the difference in arccosines is positive.

In the following lemma we compute the derivative of $\mathbb{E}[\nabla_k]$ with respect to $x$.

**Lemma E.7.**

$$\frac{d}{dx}\mathbb{E}_{U,V \sim \mathbb{P}_x}[\nabla_k] = m\frac{d}{dx}\mathbb{E}_{U,V \sim \mathbb{P}_x}\left[\frac{\partial \mathcal{L}_i}{\partial \theta_k}\right]$$

$$= \frac{-m}{1 - x^2}\theta_{k'}^T \mu_{k'} \sum_{j \neq i}\mathbb{E}_{U,V \sim \mathbb{P}_x}\left[p_{ij}p_{ii}\left(\theta_{k'}^T u_{k'}\right)\left(\mu_{k'}^T u_{k'}^{(i)} - x\mu_{k'}^T v_{k'}^{(i)}\right)\left(\frac{\partial(z_{ij} - z_{ii})}{\partial \theta_k}\right)\right].$$

We will analyze this quantity by explicitly taking the expectation with respect to some set of random variables. Let $S = \{U_k, V_k, U_{k'}, V_{k'}\}$ consist of the random variables $u_{k'}^{(i)}$, $u_k^{(i)}$, and $v_{k'}^{(i)}$, $v_k^{(i)}$ for all $i \in [m]$. Define $q_{ij}$ to be the logits when all variables in $S$ are set to 0 (Thus explicitly, $q_{ij} = \frac{\exp\left(\sum_{\tilde{k} \neq k, k'} \theta_{\tilde{k}}^T u_{\tilde{k}}^{(i)} \theta_{\tilde{k}}^T v_{\tilde{k}}^{(j)}\right)}{\sum_{j'} \exp\left(\sum_{\tilde{k} \neq k, k'} \theta_{\tilde{k}}^T u_{\tilde{k}}^{(i)} \theta_{\tilde{k}}^T v_{\tilde{k}}^{(j')}\right)}$). We will use the notation $j \sim q$ to denote the distribution on $[m]$ with mass $q_{ij}$ on $j$.

Let

$$h(S) := \left(\theta_{k'}^T u_{k'}\right) \left(\mu_{k'}^T u_{k'}^{(i)} - x\mu_{k'}^T v_{k'}^{(i)}\right) \left(\frac{\partial(z_{ij} - z_{ii})}{\partial \theta_k}\right), \tag{10}$$

and

$$h_1(S) = \left(\theta_{k'}^T u_{k'}\right) \left((1 - x^2)\mu_{k'}^T u_{k'}^{(i)}\right) 2\alpha_k \left((\mu_k^T u_k)(\theta_k^T u_k)\mu_k^T\right), \tag{11}$$

which are the terms that appear in the right hand side of Lemma E.7 after $p_{ii}p_{ij}$. Observe that

$$\mathbb{E}_S[h(S) - h_1(S)] = 0.$$

The following four lemmas serve to bound $\frac{d}{dx}\mathbb{E}_S\left[\mu_k^T \nabla_k\right]$ and $\frac{d}{dx}\mathbb{E}_S\left[\theta_k^T \nabla_k\right]$. We call the terms of the form $\mathbb{E}p_{ii}p_{ij}(h(S) - h_1(S))$ "junk" terms, and our goal will be to show that these terms are small. We will control more closely the terms of the form $\mathbb{E}p_{ii}p_{ij}(h_1(S))$.

**Lemma E.8** (Junk Terms for $\mu_k$ term.)**.** *If $\|\theta_k\| \leq 1$ and $\|\theta_{k'}\| \leq 1$, then for some universal constant $C$*

$$\left|\mathbb{E}_S\left[p_{ii}p_{ij}\mu_k^T(h(S) - h_1(S))\right]\right| \leq Cq_{ii}q_{ij}\left(\|\theta_{k'}\|^3\|\theta_k\|^3 + \|\theta_{k'}^\|\|\|\theta_k\|^3 + \alpha_k\left(\|\theta_{k'}\|^3\|\theta_k^\|\|\right)\right).$$

**Lemma E.9** (Good Term for $\mu_k$ term.)**.** *If $\|\theta_k\| \leq 1$ and $\|\theta_{k'}\| \leq 1$, then for some universal constant $C$*

$$\left|\mathbb{E}_S\left[p_{ii}p_{ij}\mu_k^T h_1(S)\right]\right| \geq 2\alpha_k(1 - x^2)q_{ii}q_{ij}\left(\|\theta_{k'}^\|\|\|\theta_k^\|\|\right)\left(1 - C(\|\theta_{k'}\|^2 + \|\theta_k\|^2)\right).$$

Plugging these two lemmas into Lemma E.7 yields the following corollary.

**Corollary E.9.1** (Total $\mu_k$ term.)**.** *If for a sufficiently large constant $C$, $|\theta_k^T \mu_k| \leq \frac{1-\alpha_{k'}^2}{C}\|\theta_k\|$, $\|\theta_{k'}\|^3 \leq |\theta_{k'}^T \mu_k|$, and $\|\theta_k\|^2 \leq \frac{\alpha_k(1-\alpha_{k'}^2)}{C}$, then*

$$(\mu_k^T\theta_k)\frac{d}{dx}\mathbb{E}_{\mathbb{P}_x}\left[\mu_k^T\nabla_k\right] \geq \frac{m}{2}\mathbb{E}_{U,V\setminus S}\left[\sum_{i,j}q_{ii}q_{ij}2\alpha_k\|\theta_{k'}^\|\|^2\|\theta_k^\|\|^2\right].$$

**Lemma E.10** (Junk Terms for $\theta_k$ term.)**.** *If $\|\theta_k\| \leq 1$ and $\|\theta_{k'}\| \leq 1$, then for some universal constant $C$*

$$\left|\mathbb{E}_S\left[p_{ii}p_{ij}\theta_k^T(h(S) - h_1(S))\right]\right| \leq Cq_{ii}q_{ij}\left(\|\theta_{k'}\|^3\|\theta_k\|^4 + \|\theta_{k'}^\|\|\|\theta_k\|^4 + \alpha_k\left(\|\theta_{k'}\|^3\|\theta_k\|\|\theta_k^\|\| + \|\theta_{k'}^\|\|\|\theta_k\|^3\|\theta_k^\|\|\right)\right).$$

**Lemma E.11** (Good Term for $\theta_k$ term.)**.** *If $\|\theta_k\| \leq 1$ and $\|\theta_{k'}\| \leq 1$, then for some universal constant $C$*

$$\left|\mathbb{E}_S\left[p_{ii}p_{ij}\theta_k^T h_1(S)\right]\right| \leq (1 - x^2)2\alpha_k q_{ii}q_{ij}\left(\|\theta_{k'}^\|\|\|\theta_k^\|\|^2\right)\left(1 + C(\|\theta_{k'}\|^2 + \|\theta_k\|^2)\right).$$

Plugging these two lemmas into Lemma E.7 yields the following corollary.

**Corollary E.11.1** (Total $\theta_k$ term.)**.** *If for a sufficiently large constant $C$, $\|\theta_k^\|\| \leq \frac{1-x^2}{C}\|\theta_k\|$, $\|\theta_{k'}\|^3 \leq \|\theta_{k'}^\|\|$, $\|\theta_k\|^2 \leq \frac{\alpha_k(1-x^2)}{C}$, then*

$$\frac{(\mu_k^T\theta_k)^2}{\|\theta_k\|^2}\left|\frac{d}{dx}\mathbb{E}_{\mathbb{P}_x}\left[\theta_k^T\nabla_k\right]\right| \leq \frac{m}{2}\mathbb{E}_{U,V\setminus S}\left[\sum_{i,j}q_{ii}q_{ij}\alpha_k\|\theta_{k'}^\|\|^2\|\theta_k^\|\|^2\right].$$

Combining Corollaries E.9.1 and E.11.1, we obtain the following lemma.

**Lemma E.12.** *If for a sufficiently large constant $C$, $\|\theta_k^\|\| \leq \frac{1-x^2}{C}\|\theta_k\|$, $\|\theta_{k'}\|^3 \leq \|\theta_{k'}^\|\|$, $\|\theta_k\|^2 \leq \frac{\alpha_k(1-x^2)}{C}$, then*

$$\mathbb{E}_{U,V\sim\mathbb{P}_x}\left[-(\mu_k^T\theta_k)(\mu_k^T\nabla_k) + \frac{\theta_k^T\nabla_k(\mu_k^T\theta_k)^2}{\|\theta_k\|_2^2}\right] < 0. \tag{12}$$

Theorem 4.1 now follows.

### E.4 PROOFS OF LEMMAS

To prove the Lemmas E.4 and E.5, we will use the following well-known formula for the moment generating function (MGF) of the half-normal distribution.

**Lemma E.13** (MGF of half-normal distribution). *The MGF of the half-normal distribution is*

$$\mathbb{E}_{X \sim \mathcal{N}(0,1)|X>0}[e^{t|X|}] = 2e^{t^2/2}\Phi(t),$$

*where $\Phi(t)$ is the cumulative distribution of a normal random variable.*

*Proof of Lemma E.4.*

$$\mathbb{E}_X\left[|X|^c \exp(t|X|)\exp(tX^2)\right] = \frac{1}{\sigma\sqrt{2\pi}}\int_{-\infty}^{\infty}|x|^c\exp(t|x|)\exp(tx^2)\exp\left(-\frac{x^2}{2\sigma^2}\right)dx$$

$$= \frac{\sqrt{1-2\sigma^2 t}}{\left(\frac{\sigma}{\sqrt{1-2\sigma^2 t}}\right)\sqrt{2\pi}}\int_{-\infty}^{\infty}|x|^c\exp(t|x|)\exp\left(-\frac{x^2}{2\left(\frac{\sigma}{\sqrt{1-2\sigma^2 t}}\right)^2}\right)dx$$

$$= \sqrt{1-2\sigma^2 t}\,\mathbb{E}_{Z\sim\mathcal{N}(0,r)|Z\geq 0}[Z^c\exp(tZ)],$$

where $r = \frac{\sigma}{\sqrt{1-2\sigma^2 t}}$. To evaluate this, we use the MGF of the half-normal distribution in Lemma E.13. Thus for some constant $C$, for all $c \in \{1, 2, 3, 4\}$,

$$\mathbb{E}_{X\sim\mathcal{N}(0,1)|X>0}\left[c!|X|^c e^{t|X|}\right] \leq \mathbb{E}_{X\sim\mathcal{N}(0,1)|X>0}\left[\frac{d^c}{dt^c}e^{t|X|}\right]$$

$$\leq C\left(1+t^c\right)e^{t^2/2}.$$

So for some constant $C$ (whose value changes throughout this equation), so long as $\sigma \leq \frac{1}{C}$,

$$\sqrt{1-2\sigma^2 t}\,\mathbb{E}_{Z\sim\mathcal{N}(0,r)|Z\geq 0}[Z^c\exp(tZ)] = \sqrt{1-2\sigma^2 t}\,\mathbb{E}_{X\sim\mathcal{N}(0,1)|Z\geq 0}[r^c Z^c\exp(rtZ)]$$

$$\leq \sqrt{1-2\sigma^2 t}\,Cr^c\left(1+(tr)^c\right)e^{(tr)^2/2}$$

$$\leq C\sigma^c.$$

This proves the first statement in the lemma. To prove the second, we first take the expectation over $X$, and using the half-Gaussian MGF as before, we obtain

$$\mathbb{E}_X\mathbb{E}_Y\left[|X|^c|Y|^d\exp(t|X|)\exp(|XY|)\right] \leq C\mathbb{E}_Y\left[|Y|^d\sigma^c(1+(t+|Y|)^c)e^{(t+|Y|)^2/2}\right]$$

Now applying the first statement to take the expectation over $Y$, we obtain

$$\mathbb{E}_Y\left[|Y|^d(1+(t+|Y|)^c)e^{(t+|Y|)^2/2}\right] \leq C\sigma^c\rho^d.$$

$\square$

*Proof of Lemma E.5.* We prove the lemma by induction on $c$. Suppose $c = 0$. Then by plugging in the MGF for the half-normal distribution from Lemma E.13, for some constant $C$, we have

$$\mathbb{E}_{X\sim\mathcal{N}(0,1)|X>0}[(e^{t|X|}-1)] = 2e^{t^2/2}\Phi(t) - 1 \tag{13}$$

$$\leq 2e^{t^2/2}\left(\frac{1+t}{2}\right) - 1 \tag{14}$$

$$\leq \left(e^{t^2/2}-1\right) + te^{t^2/2} \tag{15}$$

$$\leq Ct, \tag{16}$$

thus

$$\mathbb{E}_{X \sim \mathcal{N}(0,\sigma^2)}[(e^{t|X|} - 1)] = \mathbb{E}_{X \sim \mathcal{N}(0,\sigma^2)|X>0}[(e^{\sigma t|X|} - 1)] \leq Ct\sigma.$$

Now for $c \geq 1$, by Stein's Lemma, we have (for a new constant $C$),

$$\mathbb{E}_{X \sim \mathcal{N}(0,\sigma^2)}[|X|^c(e^{t|X|} - 1)] = \mathbb{E}_{X \sim \mathcal{N}(0,\sigma^2)}[X|X|^{c-1}\text{sign}(X)(e^{t|X|} - 1)] \tag{17}$$

$$= \sigma^2 \mathbb{E}_{X \sim \mathcal{N}(0,\sigma^2)}\left[\frac{d}{dX}\left(|X|^{c-1}\text{sign}(X)(e^{t|X|} - 1)\right)\right] \tag{18}$$

$$= \sigma^2 \mathbb{E}_{X \sim \mathcal{N}(0,\sigma^2)}\left[(c-2)\left(|X|^{c-2}(e^{t|X|} - 1)\right) + \left(|X|^{c-1}(te^{t|X|})\right)\right] \tag{19}$$

$$\leq Ct\sigma^{c+1}. \tag{20}$$

where in the last step we used the inductive hypothesis and Lemma E.4. $\square$

*Proof of Lemma E.6.* First observe that

$$\lim_{\eta \to 0} \frac{1}{\eta}\left(\mathbb{E}_{U,V}\left[\arccos\left(\frac{|\mu_k^T \theta_k^+|}{\|\theta_k^+\|_2}\right)\right] - \arccos\left(\frac{|\mu_k^T \theta_k|}{\|\theta_k\|_2}\right)\right)$$

$$= \lim_{\eta \to 0} \frac{1}{\eta}\left(\mathbb{E}_{U,V}\left[\arccos\left(\frac{|\mu_k^T(\theta_k(1 - \eta\lambda) - \eta\nabla_k)|}{\|\theta_k(1 - \eta\lambda) - \eta\nabla_k\|_2}\right)\right] - \arccos\left(\frac{|\mu_k^T \theta_k|}{\|\theta_k\|_2}\right)\right)$$

$$= \lim_{\eta \to 0} \frac{1}{\eta}\left(\mathbb{E}_{U,V}\left[\arccos\left(\frac{|\mu_k^T(\theta_k - \frac{\eta}{1-\eta\lambda}\nabla_k)|}{\|\theta_k - \frac{\eta}{1-\eta\lambda}\nabla_k\|_2}\right)\right] - \arccos\left(\frac{|\mu_k^T \theta_k|}{\|\theta_k\|_2}\right)\right)$$

$$= \mathbb{E}_{U,V}\left[\frac{d}{d\eta}\arccos\left(\frac{|\mu_k^T(\theta_k - \eta\nabla_k)|}{\|\theta_k - \eta\nabla_k\|_2}\right)(0)\right],$$

since $\lim_{\eta \to 0} \frac{\eta}{1-\eta\lambda} = 0$. Now

$$\frac{d}{d\eta}\arccos\left(\frac{|\mu_k^T(\theta_k - \eta\nabla_k)|}{\|\theta_k - \eta\nabla_k\|_2}\right)(0) = \arccos'\left(\frac{|\mu_k^T \theta_k|}{\|\theta_k\|_2}\right)\frac{d}{d\eta}\left(\frac{|\mu_k^T(\theta_k - \eta\nabla_k)|}{\|\theta_k - \eta\nabla_k\|_2}\right)(0)$$

$$= \arccos'\left(\frac{|\mu_k^T \theta_k|}{\|\theta_k\|_2}\right)\left(\frac{-\text{sign}(\mu_k^T \theta_k)\mu_k^T \nabla_k\|\theta_k\| + |\mu_k^T \theta_k|\frac{\theta_k^T \nabla_k}{\|\theta_k\|}}{\|\theta_k\|_2^2}\right)$$

$$= N\left(-\mu_k^T \theta_k \mu_k^T \nabla_k + (\mu_k^T \theta_k)^2\frac{\theta_k^T \nabla_k}{\|\theta_k\|^2}\right),$$

where $N = \arccos'\left(\frac{|\mu_k^T \theta_k|}{\|\theta_k\|_2}\right)\frac{1}{\|\theta_k\|\|\mu_k^T \theta_k|}$. The lemma follows by taking the expectation over $U, V$, and observing derivative of $\arccos(x)$ is negative whenever $x$ is positive. $\square$

*Proof of Lemma E.7.* First observe that by symmetry, we have

$$\frac{d}{dx}\mathbb{E}_{U,V \sim \mathbb{P}_x}[\nabla_k] = m\frac{d}{dx}\mathbb{E}_{U,V \sim \mathbb{P}_x}\left[\frac{\partial \mathcal{L}_i}{\partial \theta_k}\right].$$

To make this expectation easier to analyze, we express the random variable $(U(x), V(x)) \sim \mathbb{P}_x$ as an interpolation of Gaussians in the coordinate $\mu_{k'}^T v_{k'}^{(i)}$. Let $\xi \sim \mathcal{N}(0, 1)$, and define $(U, V) \sim \mathbb{P}_1$, such that $\mu_{k'}^T v_{k'}^{(i)} = \mu_{k'}^T u_{k'}^{(i)}$. For $x \in [0, 1)$, define $(U(x), V(x))$ to have

$$\mu_{k'}^T v_{k'}^{(i)}(x) = x\mu_{k'}^T u_{k'}^{(i)} + \sqrt{1 - x^2}\xi, \tag{21}$$

and otherwise be the same as $(U, V)$. It is easy to check that $(U(x), V(x)) \sim \mathbb{P}_x$.

Now

$$\frac{d}{dx}\mathbb{E}_{U,V \sim \mathbb{P}_x}\left[\frac{\partial \mathcal{L}_i(\Theta; U, V)}{\partial \theta_k}\right] = \mathbb{E}_{U,V \sim \mathbb{P}_1, \xi}\left[\frac{d}{dx}\frac{\partial \mathcal{L}_i(\Theta; U(x), V(x))}{\partial \theta_k}\right].$$

Taking the derivative of the cross-entropy loss, we have

$$\frac{d}{dx}\frac{\partial \mathcal{L}_i(\Theta; U(x), V(x))}{\partial \theta_k} = \frac{d}{dx}\left(\sum_{j \neq i} p_{ij}\left(\frac{\partial(z_{ij} - z_{ii})}{\partial \theta_k}\right)\right)$$

$$= \sum_{j \neq i}\frac{dp_{ij}}{d\mu_{k'}^T v_{k'}^{(i)}(x)}\frac{d\mu_{k'}^T v_{k'}^{(i)}(x)}{dx}\frac{\partial(z_{ij} - z_{ii})}{\partial \theta_k}$$

$$= \sum_{j \neq i} -p_{ij}p_{ii}\frac{dz_{ii}}{d\mu_{k'}^T v_{k'}^{(i)}(x)}\left(\mu_{k'}^T u_{k'}^{(i)} - \frac{x}{\sqrt{1 - x^2}}\xi\right)\left(\frac{\partial(z_{ij} - z_{ii})}{\partial \theta_k}\right)$$

where the variables $z_{ij}$ and $p_{ij}$ are the similarity scores and the softmaxes from the data $(U(x), V(x))$. Here the first line is by Lemma E.2, and the second line holds by chain rule since $\frac{\partial z_{ij}}{\partial \theta_k} - \frac{\partial z_{ii}}{\partial \theta_k}$ does not depend on $v_{k'}^{(i)}$. The third line uses the proof of Claim E.14 to take the derivative of $p_{ij}$, and Equation 21 to take the derivative of $\mu_{k'}^T v_{k'}^{(i)}(x)$.

Now we reparameterize $\mu_{k'}^T u_{k'}^{(i)} - \frac{x}{\sqrt{1-x^2}}\xi$ as follows:

$$\mu_{k'}^T u_{k'}^{(i)} - \frac{x}{\sqrt{1 - x^2}}\xi = \left(\frac{1}{1 - x^2}\right)\mu_{k'}^T u_{k'}^{(i)} - \frac{x}{1 - x^2}\mu_{k'}^T v_{k'}^{(i)}(x).$$

Plugging in this reparameterization and $\frac{dz_{ii}}{d\mu_{k'}^T v_{k'}^{(i)}(x)} = \theta_{k'}^T \mu_{k'}\theta_{k'}^T u_{k'}$, we obtain

$$\frac{d}{dx}\mathbb{E}_{U,V \sim \mathbb{P}_x}\left[\frac{\partial \mathcal{L}_i(\Theta; U, V)}{\partial \theta_k}\right] = \frac{-1}{1 - x^2}\sum_{j \neq i}\mathbb{E}_{U,V \sim \mathbb{P}_x}\left[p_{ij}p_{ii}\left(\theta_{k'}^T \mu_{k'}\theta_{k'}^T u_{k'}\right)\left(\mu_{k'}^T u_{k'}^{(i)} - x\mu_{k'}^T v_{k'}^{(i)}\right)\left(\frac{\partial(z_{ij} - z_{ii})}{\partial \theta_k}\right)\right].$$

$\square$

We now prove Lemmas E.8, E.9, E.10, and E.11.

**Notation.** Since $i$ is fixed throughout, we drop the $(i)$ superscripts and let $u_k = u_k^{(i)}$ and $v_k = v_k^{(i)}$. We will introduce the following random variables, which are all independent, to simplify the exposition:

- $\xi_j := \theta_k^T v_k^{(j)}$ for $j \neq i$. Thus $\xi_j \sim \mathcal{N}(0, \|\theta_k\|^2)$.
- $\xi_j' := \theta_{k'}^T v_{k'}^{(j)}$ for $j \neq i$. Thus $\xi_j' \sim \mathcal{N}(0, \|\theta_{k'}\|^2)$.

- $\xi_i := (\theta_k^\perp)^T v_k + (\theta_k^\parallel)^T (v_k - \alpha_k u_k)$. Thus $\xi_i \sim \mathcal{N}(0, \|\theta_k^\perp\|^2 + (1 - \alpha_k^2)\|\theta_k^\parallel\|^2)$.
- $\xi_i' := (\theta_{k'}^\perp)^T v_{k'}$. Thus $\xi_i' \sim \mathcal{N}(0, \|\theta_{k'}^\perp\|^2 \|\theta_{k'}^\parallel\|^2)$.
- $\zeta_i' := (\theta_{k'}^\parallel)^T (v_{k'} - \alpha_{k'} u_{k'})$. Thus $\zeta_i' \sim \mathcal{N}(0, (1 - \alpha_{k'}^2)\|\theta_{k'}^\parallel\|^2)$.
- $y = (\theta_k^\parallel)^T u_k$. Thus $y \sim \mathcal{N}(0, \|\theta_k^\parallel\|^2)$.
- $y' = (\theta_{k'}^\parallel)^T u_{k'}$. Thus $y' \sim \mathcal{N}(0, \|\theta_{k'}^\parallel\|^2)$.
- $\eta_i := (\theta_k^\perp)^T u_k$. Thus $\eta_i \sim \mathcal{N}(0, \|\theta_k^\perp\|^2)$.
- $\eta_i' := (\theta_{k'}^\perp)^T u_{k'}$. Thus $\eta_i' \sim \mathcal{N}(0, \|\theta_{k'}^\perp\|^2)$.

For any such random variable $X$, we use $\sigma_X^2$ to denote its variance. Observe that

$$\frac{p_{ii}p_{ij}}{q_{ii}q_{ij}} = \frac{\exp\left(\theta_k^T u_k \theta_k^T v_k\right)\exp\left(\theta_{k'}^T u_{k'}\theta_{k'}^T v_{k'}\right)}{\mathbb{E}_{j'\sim q}\exp\left(\theta_k^T u_k \theta_k^T v_k^{(j')}\right)\exp\left(\theta_{k'}^T u_{k'}\theta_{k'}^T v_{k'}^{(j')}\right)} \frac{\exp\left(\theta_k^T u_k \theta_k^T v_k^{(j)}\right)\exp\left(\theta_{k'}^T u_{k'}\theta_{k'}^T v_{k'}^{(j)}\right)}{\mathbb{E}_{j'\sim q}\exp\left(\theta_k^T u_k \theta_k^T v_k^{(j')}\right)\exp\left(\theta_{k'}^T u_{k'}\theta_{k'}^T v_{k'}^{(j')}\right)}.$$

We will use the following two claims in the proofs of all four lemmas.

**Claim E.14.** *For $\beta \in \{\xi_j, \xi_j', \xi_i, \xi_i', \zeta_i', \eta_i, \eta_i', x, x'\}$, let $\bar{\beta}_{j'} := \frac{\partial}{\partial\beta}\left(\theta_k^T u_k \theta_k^T v_k^{(j')} + \theta_{k'}^T u_{k'}\theta_{k'}^T v_{k'}^{(j')}\right)$. Then*

$$\left|\frac{\partial p_{ii}p_{ij}}{\partial\beta}\right| \le p_{ii}p_{ij}\left(|\bar{\beta}_j| + |\bar{\beta}_i| + 2\mathbb{E}_{j'\sim q}|\bar{\beta}_{j'}|\right).$$

*If additionally $\gamma \in \{\xi_j, \xi_j', \xi_i, \xi_i', \zeta_i', \eta_i, \eta_i'\}$ and $\gamma \perp \{\bar{\beta}_{j'}\}_{j'\in[m]}$, then*

$$\left|\frac{\partial}{\partial\gamma}\frac{\partial p_{ii}p_{ij}}{\partial\beta}\right| \le p_{ii}p_{ij}\left(\left(|\bar{\beta}_j| + |\bar{\beta}_i| + 2\mathbb{E}_{j'\sim q}|\bar{\beta}_{j'}|\right)\left(|\bar{\gamma}_j| + |\bar{\gamma}_i| + 2\mathbb{E}_{j'\sim q}|\bar{\gamma}_{j'}|\right) + 2\mathbb{E}_{j'\sim q}|\bar{\beta}_{j'}\bar{\gamma}_{j'}| + 2(\mathbb{E}_{j'\sim q}|\bar{\beta}_{j'}|)(\mathbb{E}_{j'\sim q}|\bar{\gamma}_{j'}|)\right).$$

*Proof.* By a straightforward quotient-rule computation of the derivative of $\frac{p_{ij}}{q_{ij}}$, recalling that $q_{ij}$ is independent of $S$, we obtain

$$\frac{\partial p_{ij}}{\partial\beta} = p_{ij}\left(\bar{\beta}_j - \mathbb{E}_{j'\sim q}\bar{\beta}_{j'}p_{ij'}\right).$$

By applying product to the expression above, we obtain

$$\frac{\partial p_{ii}p_{ij}}{\partial\beta} = p_{ii}p_{ij}\left(\bar{\beta}_j + \bar{\beta}_i - 2\mathbb{E}_{j'\sim q}\bar{\beta}_{j'}p_{ij'}\right).$$

Taking absolute values and using the fact that $p_{ij'} \le 1$, we obtain the first result.

Next we take the derivative of $p_{ij}$ with respect to both $\beta$ and $\gamma$. Using the expression above for $\frac{\partial p_{ij}}{\partial\beta}$, we obtain

$$\frac{\partial}{\partial\gamma}\frac{\partial p_{ij}}{\partial\beta} = p_{ij}\left(\left(\bar{\beta}_j - \mathbb{E}_{j'\sim q}\bar{\beta}_{j'}p_{ij'}\right)\left(\bar{\gamma}_j - \mathbb{E}_{j'\sim q}\bar{\gamma}_{j'}p_{ij'}\right) - \mathbb{E}_{j'\sim q}\bar{\beta}_{j'}\bar{\gamma}_{j'}p_{ij'} + (\mathbb{E}_{j'\sim q}\bar{\beta}_{j'}p_{ij'})(\mathbb{E}_{j'\sim q}\bar{\gamma}_{j'}p_{ij'})\right),$$

and

$$\frac{\partial}{\partial\gamma}\frac{\partial p_{ii}p_{ij}}{\partial\beta} = p_{ii}p_{ij}\left(\left(\bar{\beta}_j + \bar{\beta}_i - 2\mathbb{E}_{j'\sim q}\bar{\beta}_{j'}p_{ij'}\right)\left(\bar{\gamma}_j + \bar{\gamma}_i - 2\mathbb{E}_{j'\sim q}\bar{\gamma}_{j'}p_{ij'}\right) - 2\mathbb{E}_{j'\sim q}\bar{\beta}_{j'}\bar{\gamma}_{j'}p_{ij'} + 2(\mathbb{E}_{j'\sim q}\bar{\beta}_{j'}p_{ij'})(\mathbb{E}_{j'\sim q}\bar{\gamma}_{j'}p_{ij'})\right)$$

The second result follows by taking absolute values and the fact that $p_{ij'} \le 1$. $\qquad\square$

**Claim E.15.**

$$\frac{p_{ij}}{q_{ij}} \leq \exp\left(|\theta_k^T u_k \theta_k^T v_k^{(j)}|\right) \exp\left(|\theta_{k'}^T u_{k'} \theta_{k'}^T v_{k'}^{(j)}|\right) \mathbb{E}_{j'\sim q}\left[\exp\left(|\theta_k^T u_k \theta_k^T v_k^{(j')}|\right) \exp\left(|\theta_{k'}^T u_{k'} \theta_{k'}^T v_{k'}^{(j')}|\right)\right].$$

*Proof.* This follows directly from using Jenson's inequality on the distribution $j' \sim q$ to show that

$$\frac{1}{\mathbb{E}_{j'\sim q}\left[\exp\left(\theta_k^T u_k \theta_k^T v_k^{(j')}\right) \exp\left(\theta_{k'}^T u_{k'} \theta_{k'}^T v_{k'}^{(j')}\right)\right]} \leq \mathbb{E}_{j'\sim q}\left[\exp\left(-\theta_k^T u_k \theta_k^T v_k^{(j')}\right) \exp\left(-\theta_{k'}^T u_{k'} \theta_{k'}^T v_{k'}^{(j')}\right)\right]$$

$$\leq \mathbb{E}_{j'\sim q}\left[\exp\left(|\theta_k^T u_k \theta_k^T v_k^{(j')}|\right) \exp\left(|\theta_{k'}^T u_{k'} \theta_{k'}^T v_{k'}^{(j')}|\right)\right].$$

$\square$

**Claim E.16.**

$$\left|1 - \frac{p_{ij}}{q_{ij}}\right| \leq Z_j - 1,$$

*where* $Z_j := \exp\left(|\theta_k^T u_k \theta_k^T v_k^{(j)}|\right) \exp\left(|\theta_{k'}^T u_{k'} \theta_{k'}^T v_{k'}^{(j)}|\right) \mathbb{E}_{j'\sim q}\left[\exp\left(|\theta_k^T u_k \theta_k^T v_k^{(j')}|\right) \exp\left(|\theta_{k'}^T u_{k'} \theta_{k'}^T v_{k'}^{(j')}|\right)\right].$

*Proof.* Note that for any $x \geq 0$, we have $|1 - x| \leq \max\left(x - 1, \frac{1}{x} - 1\right)$. By Claim E.15, $\frac{p_{ij}}{q_{ij}} - 1$ is at most the desired value given in this claim.

Now

$$\frac{q_{ij}}{p_{ij}} = \frac{\mathbb{E}_{j'\sim q}\left[\exp\left(\theta_k^T u_k \theta_k^T v_k^{(j')}\right) \exp\left(\theta_{k'}^T u_{k'} \theta_{k'}^T v_{k'}^{(j')}\right)\right]}{\exp\left(\theta_k^T u_k \theta_k^T v_k^{(j)}\right) \exp\left(\theta_{k'}^T u_{k'} \theta_{k'}^T v_{k'}^{(j)}\right)}$$

$$\leq \exp\left(|\theta_k^T u_k \theta_k^T v_k^{(j)}|\right) \exp\left(|\theta_{k'}^T u_{k'} \theta_{k'}^T v_{k'}^{(j)}|\right) \mathbb{E}_{j'\sim q}\left[\exp\left(|\theta_k^T u_k \theta_k^T v_k^{(j')}|\right) \exp\left(|\theta_{k'}^T u_{k'} \theta_{k'}^T v_{k'}^{(j')}|\right)\right].$$

This yields the claim. $\square$

*Proof of Lemma E.8.* Expanding $h(S) - h_1(S)$, we see that we need to control the following terms:

1. (a) $\left|\mathbb{E}_S\left[p_{ii}p_{ij}\left(\eta_i'\left(\mu_{k'}^T u_{k'} - x\mu_{k'}^T v_{k'}\right)\right)\left(\mu_k^T u_k \xi_j\right)\right]\right|$, (b) $\left|\mathbb{E}_S\left[p_{ii}p_{ij}\left(y'\left(\mu_{k'}^T u_{k'} - x\mu_{k'}^T v_{k'}\right)\right)\left(\mu_k^T u_k \xi_j\right)\right]\right|$

2. (a) $\left|\mathbb{E}_S\left[p_{ii}p_{ij}\left(\eta_i'\left(\mu_{k'}^T u_{k'} - x\mu_{k'}^T v_{k'}\right)\right)\left(\mu_k^T u_k \xi_i\right)\right]\right|$, (b) $\left|\mathbb{E}_S\left[p_{ii}p_{ij}\left((y'\left(\mu_{k'}^T u_{k'} - x\mu_{k'}^T v_{k'}\right)\right)\left(\mu_k^T u_k \xi_i\right)\right]\right|$

3. (a) $\left|\alpha_k\mathbb{E}_S\left[p_{ii}p_{ij}\left(\eta_i'\left(\mu_{k'}^T u_{k'} - x\mu_{k'}^T v_{k'}\right)\right)\left(\mu_k^T u_k y\right)\right]\right|$, (b) $\left|\alpha_k\mathbb{E}_S\left[p_{ii}p_{ij}\left(y'(-x\xi_i')\right)\left(\mu_k^T u_k y\right)\right]\right|$

4. (a) $\left|\mathbb{E}_S\left[p_{ii}p_{ij}\left(\eta_i'\left(\mu_{k'}^T u_{k'} - x\mu_{k'}^T v_{k'}\right)\right)\left(\xi_i(v_k - v_k^{(j)})^T \mu_k\right)\right]\right|$
   (b) $\left|\mathbb{E}_S\left[p_{ii}p_{ij}\left(y'\left(\mu_{k'}^T u_{k'} - x\mu_{k'}^T v_{k'}\right)\right)\left(\xi_i(v_k - v_k^{(j)})^T \mu_k\right)\right]\right|$

5. (a) $\left|\alpha_k\mathbb{E}_S\left[p_{ii}p_{ij}\left(\eta_i'\left(\mu_{k'}^T u_{k'} - x\mu_{k'}^T v_{k'}\right)\right)\left(y(v_k - v_k^{(j)})^T \mu_k\right)\right]\right|$
   (b) $\left|\mathbb{E}_S\left[p_{ii}p_{ij}\left(y'\left(\mu_{k'}^T u_{k'} - x\mu_{k'}^T v_{k'}\right)\right)\left(y(v_k - \alpha_k u_k - v_k^{(j)})^T \mu_k\right)\right]\right|$

We begin by bounding the terms where the expression after $p_{ii}p_{ij}$ has two independent mean-0 terms, mainly (1a), (2a), (4a). The first step is to apply Stein's Lemma (Lemma E.3) twice to these two terms, which we will call $\beta$ and $\gamma$. Let $\beta\gamma g(S \setminus \{\beta, \gamma\})$ be the terms after $p_{ii}p_{ij}$. Then we have

$$|\mathbb{E}_S[p_{ii}p_{ij}\beta\gamma g(S \setminus \{\beta, \gamma\})]| \leq \sigma_\beta^2 \sigma_\gamma^2 \left|\mathbb{E}_S\left[\left|\frac{\partial}{\partial \gamma}\frac{\partial p_{ii}p_{ij}}{\partial \beta}\right| |g(S \setminus \{\beta, \gamma\})|\right]\right|.$$

Next we apply the final result in Claim E.14 to bound the absolute value of $\left|\frac{\partial}{\partial \gamma}\frac{\partial p_{ii}p_{ij}}{\partial \beta}\right|$. Once we do this, we achieve

$$|\mathbb{E}_S[p_{ii}p_{ij}\beta\gamma g(S \setminus \{\beta, \gamma\})]| \leq \sigma_\beta^2 \sigma_\gamma^2 q_{ii}q_{ij}\mathbb{E}_S\left[Z|g(S \setminus \{\beta, \gamma\})| \sum_{j',\ell\in[m]} c_{j',\ell}|\bar{\beta_{j'}}||\bar{\gamma_\ell}|\right],$$

where $\sum_{j',\ell\in[m]} c_{j',\ell} \leq C$ for some constant $C$, and $Z := \frac{p_{ii}p_{ij}}{q_{ii}q_{ij}}$. Finally, we use the bound on $Z$ from Claim E.15, and then Lemma E.4 to take the expectation over $S$, iteratively applying Lemma E.4 to each variable in $S$. Thus we have, for some (different) constant $C$,

1. $\left|\mathbb{E}_S\left[p_{ii}p_{ij}\left(\eta_i'\left(\mu_{k'}^T u_{k'} - x\mu_{k'}^T v_{k'}\right)\right)\left(\mu_k^T u_k \xi_j\right)\right]\right| \leq Cq_{ii}q_{ij}\sigma_{\eta_i'}^2 \sigma_{\xi_j}^2 \|\theta_{k'}\|\|\theta_k\| = Cq_{ii}q_{ij}\|\theta_{k'}^\perp\|^2\|\theta_{k'}\|\|\theta_k\|^3 \leq Cq_{ii}q_{ij}\|\theta_{k'}\|^3\|\theta_k\|^3.$

2. $\left|\mathbb{E}_S\left[p_{ii}p_{ij}\left(\eta_i'\left(\mu_{k'}^T u_{k'} - x\mu_{k'}^T v_{k'}\right)\right)\left(\mu_k^T u_k \xi_i\right)\right]\right| \leq Cq_{ii}q_{ij}\sigma_{\eta_i'}^2 \sigma_{\xi_i}^2 \|\theta_{k'}\|\|\theta_k\| \leq Cq_{ii}q_{ij}\|\theta_{k'}^\perp\|^2\|\theta_{k'}\|\|\theta_k\|^3 \leq Cq_{ii}q_{ij}\|\theta_{k'}\|^3\|\theta_k\|^3.$

3. $\left|\mathbb{E}_S\left[p_{ii}p_{ij}\left(\eta_i'\left(\mu_{k'}^T u_{k'} - x\mu_{k'}^T v_{k'}\right)\right)\left(\xi_i(v_k - v_k^{(j)})^T\mu_k\right)\right]\right| \leq Cq_{ii}q_{ij}\sigma_{\eta_i'}^2 \sigma_{\xi_i}^2 \|\theta_{k'}\|\|\theta_k\| \leq Cq_{ii}q_{ij}\|\theta_{k'}^\perp\|^2\|\theta_{k'}\|\|\theta_k\|^3 \leq Cq_{ii}q_{ij}\|\theta_{k'}\|^3\|\theta_k\|^3.$

Now we consider the remaining 7 terms. Here we decompose the expression inside the expectation as $p_{ii}p_{ij}\beta g(S \setminus \beta)$, where $\beta \in S$. We proceed as before, but we only apply Stein's Lemma once, to $\beta$. Applying Steins, the expression for $\frac{\partial p_{ii}p_{ij}}{\partial \beta}$ given in the first result of Claim E.14, we obtain

$$|\mathbb{E}_S[p_{ii}p_{ij}\beta g(S \setminus \beta)]| \leq \sigma_\beta^2 \left|\mathbb{E}_S\left[\left|\frac{\partial p_{ii}p_{ij}}{\partial \beta}\right| |g(S \setminus \beta)|\right]\right| \leq \sigma_\beta^2 q_{ii}q_{ij}\mathbb{E}_S\left[Z|g(S \setminus \beta)| \sum_{j'\in[m]} c_{j'}|\bar{\beta_{j'}}|\right],$$

(22)

where $\sum_{j'\in[m]} c_{j'} \leq C$ for some constant $C$, and $Z := \frac{p_{ii}p_{ij}}{q_{ii}q_{ij}}$. Finally, we plug in a bound for $Z$ in Claim E.15, an use Lemma E.4 to take the expectation over $S$, again iteratively over each variable.

Thus we have, for some (different) constant $C$,

1. $\left|\mathbb{E}_S\left[p_{ii}p_{ij}\left(y'\left(\mu_{k'}^T u_{k'} - x\mu_{k'}^T v_{k'}\right)\right)\left(\mu_k^T u_k \xi_j\right)\right]\right| \leq Cq_{ii}q_{ij}\sigma_{\xi_j}^2 \|\theta_k\|\|\theta_{k'}^\|\| = Cq_{ii}q_{ij}\|\theta_k\|^3\|\theta_{k'}^\|\|.$

2. $\left|\mathbb{E}_S\left[p_{ii}p_{ij}\left(y'\left(\mu_{k'}^T u_{k'} - x\mu_{k'}^T v_{k'}\right)\right)\left(\mu_k^T u_k \xi_i\right)\right]\right| \leq Cq_{ii}q_{ij}\sigma_{\xi_i}^2 \|\theta_k\|\|\theta_{k'}^\|\| \leq Cq_{ii}q_{ij}\|\theta_k\|^3\|\theta_{k'}^\|\|.$

3. $\left|\alpha_k\mathbb{E}_S\left[p_{ii}p_{ij}\left(\eta_i'\left(\mu_{k'}^T u_{k'} - x\mu_{k'}^T v_{k'}\right)\right)\left(\mu_k^T u_k y\right)\right]\right| \leq C\alpha_k q_{ii}q_{ij}\sigma_{\eta_i'}^2 \|\theta_{k'}\|\|\theta_k^\|\| = C\alpha_k q_{ii}q_{ij}\|\theta_{k'}^\perp\|^2\|\theta_{k'}\|\|\theta_k^\|\| \leq C\alpha_k q_{ii}q_{ij}\|\theta_{k'}\|^3\|\theta_k^\|\|.$

4. $\left|\alpha_k\mathbb{E}_S\left[p_{ii}p_{ij}\left(y'(-x\zeta_i')\right)\left(\mu_k^T u_k y\right)\right]\right| \leq C\alpha_k q_{ii}q_{ij}\sigma_{\zeta_i'}^2 \|\theta_{k'}\|\|\theta_k^\|\| = C\alpha_k q_{ii}q_{ij}\|\theta_{k'}^\|\|^2\|\theta_{k'}\|\|\theta_k^\|\|.$

5. $\left| \mathbb{E}_S \left[ p_{ii} p_{ij} \left( y' \left( \mu_{k'}^T u_{k'} - x \mu_{k'}^T v_{k'} \right) \right) \left( \xi_i (v_k - v_k^{(j)})^T \mu_k \right) \right] \right| \quad \leq \quad C q_{ii} q_{ij} \sigma_{\xi_i}^2 \|\theta_k\| \|\theta_{k'}^{\|}\| \quad \leq$
   $C q_{ii} q_{ij} \|\theta_k\|^3 \|\theta_{k'}^{\|}\|.$

6. $\left| \alpha_k \mathbb{E}_S \left[ p_{ii} p_{ij} \left( \eta_i' \left( \mu_{k'}^T u_{k'} - x \mu_{k'}^T v_{k'} \right) \right) \left( x (v_k - v_k^{(j)})^T \mu_k \right) \right] \right| \quad \leq \quad C \alpha_k q_{ii} q_{ij} \sigma_{\eta_i'}^2 \|\theta_{k'}^{\|}\| \|\theta_k^{\|}\| \quad =$
   $C \alpha_k q_{ii} q_{ij} \|\theta_{k'}^{\perp}\|^2 \|\theta_{k'}^{\|}\| \|\theta_k^{\|}\| \leq C \alpha_k q_{ii} q_{ij} \|\theta_{k'}\|^3 \|\theta_k^{\|}\|.$

7. $\left| \mathbb{E}_S \left[ p_{ii} p_{ij} \left( y' \left( \mu_{k'}^T u_{k'} - x \mu_{k'}^T v_{k'} \right) \right) \left( x (v_k - \alpha_k u_k - v_k^{(j)})^T \mu_k \right) \right] \right| \quad \leq \quad C q_{ii} q_{ij} \sigma_x^2 \|\theta_k\| \|\theta_{k'}^{\|}\| \quad =$
   $C q_{ii} q_{ij} \|\theta_k^{\|}\|^2 \|\theta_k\| \|\theta_{k'}^{\|}\|.$

Combining the bounds on these 10 terms proves the lemma:

$$\left| \mathbb{E}_S \left[ p_{ii} p_{ij} \mu_k^T (h(S) - h_1(S)) \right] \right| \leq C q_{ii} q_{ij} \left( \|\theta_{k'}\|^3 \|\theta_k\|^3 + \|\theta_{k'}^{\|}\| \|\theta_k\|^3 + \alpha_k \left( \|\theta_{k'}\|^3 \|\theta_k^{\|}\| \right) \right).$$

$\square$

*Proof of Lemma E.10.* The proof of Lemma E.10 is nearly identical, besides some differences in the terms we need to bound. We list them below:

1. (a) $\left| \mathbb{E}_S \left[ p_{ii} p_{ij} \left( \eta_i' \left( \mu_{k'}^T u_{k'} - x \mu_{k'}^T v_{k'} \right) \right) \left( \theta_k^T u_k \xi_j \right) \right] \right|$ (b) $\left| \mathbb{E}_S \left[ p_{ii} p_{ij} \left( y' \left( \mu_{k'}^T u_{k'} - x \mu_{k'}^T v_{k'} \right) \right) \left( \theta_k^T u_k \xi_j \right) \right] \right|$

2. (a) $\left| \mathbb{E}_S \left[ p_{ii} p_{ij} \left( \eta_i' \left( \mu_{k'}^T u_{k'} - x \mu_{k'}^T v_{k'} \right) \right) \left( \theta_k^T u_k \xi_i \right) \right] \right|$ (b) $\left| \mathbb{E}_S \left[ p_{ii} p_{ij} \left( \left( y' \left( \mu_{k'}^T u_{k'} - x \mu_{k'}^T v_{k'} \right) \right) \left( \theta_k^T u_k \xi_i \right) \right] \right|$

3. (a) $\left| \alpha_k \mathbb{E}_S \left[ p_{ii} p_{ij} \left( \eta_i' \left( \mu_{k'}^T u_{k'} - x \mu_{k'}^T v_{k'} \right) \right) \left( \theta_k^T u_k y \right) \right] \right|$ (b) $\left| \alpha_k \mathbb{E}_S \left[ p_{ii} p_{ij} \left( y' \left( \mu_{k'}^T u_{k'} - x \mu_{k'}^T v_{k'} \right) \right) \left( \eta_i y \right) \right] \right|$

4. $\left| \alpha_k \mathbb{E}_S \left[ p_{ii} p_{ij} \left( y' \left( -x \zeta_i' \right) \right) \left( \theta_k^T u_k y \right) \right] \right|$

We use the same approach as before. For the terms (1a) and (2a) we apply Stein's Lemma to $(\eta_i', \xi_j)$ and $(\eta_i', \xi_i)$ respectively. For (1b), (2b), (3a) and (3b) and (4), we apply Stein's Lemma to $\xi_j$, $\xi_i$, $\eta_i'$, $\eta_i$, and $\xi_i'$ respectively. Using Claim E.15 and then Lemma E.4 as before, we obtain the following result:

1. $\left| \mathbb{E}_S \left[ p_{ii} p_{ij} \left( \eta_i' \left( \mu_{k'}^T u_{k'} - x \mu_{k'}^T v_{k'} \right) \right) \left( \theta_k^T u_k \xi_j \right) \right] \right| \quad \leq \quad C q_{ii} q_{ij} \sigma_{\eta_i'}^2 \sigma_{\xi_j}^2 \|\theta_{k'}\| \|\theta_k\| \|\theta_k\| \quad =$
   $C q_{ii} q_{ij} \|\theta_{k'}^{\perp}\|^2 \|\theta_{k'}\| \|\theta_k\|^4 \leq C q_{ii} q_{ij} \|\theta_{k'}\|^3 \|\theta_k\|^4.$

2. $\left| \mathbb{E}_S \left[ p_{ii} p_{ij} \left( \eta_i' \left( \mu_{k'}^T u_{k'} - x \mu_{k'}^T v_{k'} \right) \right) \left( \theta_k^T u_k \xi_i \right) \right] \right| \quad \leq \quad C q_{ii} q_{ij} \sigma_{\eta_i'}^2 \sigma_{\xi_i}^2 \|\theta_{k'}\| \|\theta_k\| \|\theta_k\| \quad \leq$
   $C q_{ii} q_{ij} \|\theta_{k'}^{\perp}\|^2 \|\theta_{k'}\| \|\theta_k\|^4 \leq C q_{ii} q_{ij} \|\theta_{k'}\|^3 \|\theta_k\|^4.$

3. $\left| \mathbb{E}_S \left[ p_{ii} p_{ij} \left( y' \left( \mu_{k'}^T u_{k'} - x \mu_{k'}^T v_{k'} \right) \right) \left( \theta_k^T u_k \xi_j \right) \right] \right| \quad \leq \quad C q_{ii} q_{ij} \sigma_{\xi_j}^2 \|\theta_k\| \|\theta_k\| \|\theta_{k'}^{\|}\| \quad =$
   $C q_{ii} q_{ij} \|\theta_k\|^4 \|\theta_{k'}^{\|}\|$

4. $\left| \mathbb{E}_S \left[ p_{ii} p_{ij} \left( y' \left( \mu_{k'}^T u_{k'} - x \mu_{k'}^T v_{k'} \right) \right) \left( \theta_k^T u_k \xi_i \right) \right] \right| \quad \leq \quad C q_{ii} q_{ij} \sigma_{\xi_i}^2 \|\theta_k\| \|\theta_k\| \|\theta_{k'}^{\|}\| \quad \leq$
   $C q_{ii} q_{ij} \|\theta_k\|^4 \|\theta_{k'}^{\|}\|$

5. $\left| \alpha_k \mathbb{E}_S \left[ p_{ii} p_{ij} \left( \eta_i' \left( \mu_{k'}^T u_{k'} - x \mu_{k'}^T v_{k'} \right) \right) \left( \theta_k^T u_k y \right) \right] \right| \quad \leq \quad C \alpha_k q_{ii} q_{ij} \sigma_{\eta_i'}^2 \|\theta_{k'}\| \|\theta_k\| \|\theta_k^{\|}\| \quad =$
   $C \alpha_k q_{ii} q_{ij} \|\theta_{k'}^{\perp}\|^2 \|\theta_{k'}\| \|\theta_k\| \|\theta_k^{\|}\|$

6. $\left|\alpha_k \mathbb{E}_S\left[p_{ii}p_{ij}\left(y'\left(\mu_{k'}^T u_{k'} - x\mu_{k'}^T v_{k'}\right)\right)(\eta_i y)\right]\right| \quad \leq \quad C\alpha_k q_{ii}q_{ij}\sigma_{\eta_i}^2 \|\theta_k\|\|\theta_{k'}^{\|}\|\|\theta_k^{\|}\| \quad =$
$C\alpha_k q_{ii}q_{ij}\|\theta_k^{\perp}\|^2\|\theta_k\|\|\theta_{k'}^{\|}\|\|\theta_k^{\|}\|.$

7. $\left|\alpha_k \mathbb{E}_S\left[p_{ii}p_{ij}\left(y'\left(-x\zeta_i'\right)\right)\left(\theta_k^T u_k y\right)\right]\right| \quad \leq \quad C\alpha_k q_{ii}q_{ij}\sigma_{\zeta_i'}^2\|\theta_{k'}\|\|\theta_k\|\|\theta_k^{\|}\| \quad \leq$
$C\alpha_k q_{ii}q_{ij}\|\theta_{k'}^{\|}\|^2\|\theta_{k'}\|\|\theta_k\|\|\theta_k^{\|}\|.$

Combining the bounds on these 7 terms, proves the lemma:

$$\left|\mathbb{E}_S\left[p_{ii}p_{ij}\theta_k^T(h(S) - h_1(S))\right]\right| \leq Cq_{ii}q_{ij}\left(\|\theta_{k'}\|^3\|\theta_k\|^4 + \|\theta_{k'}^{\|}\|\|\theta_k\|^4 + \alpha_k\left(\|\theta_{k'}\|^3\|\theta_k\|\|\theta_k^{\|}\| + \|\theta_{k'}^{\|}\|\|\theta_k\|^3\|\theta_k^{\|}\|\right)\right).$$

$\square$

We now prove the lemmas on the non-junk terms.

*Proof of Lemma E.9.*

$\mathbb{E}_S\left[p_{ii}p_{ij}\left((\theta_{k'}^{\|})^T u_{k'} u_{k'}^T \mu_{k'}\right)\left(2\mu_k^T u_k \alpha_k(\theta_k^{\|})^T u_k\right)\right]$

$= \mathbb{E}_S\left[q_{ii}q_{ij}\left((\theta_{k'}^{\|})^T u_{k'} u_{k'}^T \mu_{k'}\right)\left(2\mu_k^T u_k \alpha_k(\theta_k^{\|})^T u_k\right)\right] + \mathbb{E}_S\left[(p_{ii}p_{ij} - q_{ii}q_{ij})\left((\theta_{k'}^{\|})^T u_{k'} u_{k'}^T \mu_{k'}\right)\left(2\mu_k^T u_k \alpha_k(\theta_k^{\|})^T u_k\right)\right]$

$= 2\alpha_k q_{ii}q_{ij}\theta_{k'}^T \mu_{k'}\theta_k^T \mu_k + 2\alpha_k q_{ii}q_{ij}\mathbb{E}_S\left[\left(\frac{p_{ii}p_{ij}}{q_{ii}q_{ij}} - 1\right)\left((\theta_{k'}^{\|})^T u_{k'} u_{k'}^T \mu_{k'}\right)\left(\mu_k^T u_k(\theta_k^{\|})^T u_k\right)\right].$

Now by Claim E.16, we have $\left|\frac{p_{ii}p_{ij}}{q_{ii}q_{ij}} - 1\right| \leq Z_i Z_j - 1$ (where the variable's $Z_i, Z_j$ are defined in the Claim E.16) so

$\left|\mathbb{E}_S\left[\left(\frac{p_{ii}p_{ij}}{q_{ii}q_{ij}} - 1\right)\left((\theta_{k'}^{\|})^T u_{k'} u_{k'}^T \mu_{k'}\right)\left(\mu_k^T u_k(\theta_k^{\|})^T u_k\right)\right]\right| \leq \mathbb{E}_S\left[(Z_i Z_j - 1)\left|(\theta_{k'}^{\|})^T u_{k'} u_{k'}^T \mu_{k'}\right|\left|\mu_k^T u_k(\theta_k^{\|})^T u_k\right|\right]$

$\leq C\left(\|\theta_k\|^2 + \|\theta_{k'}\|^2\right)\|\theta_{k'}^{\|}\|\|\theta_k^{\|}\|.$

Here the second inequality follows from applying Lemma E.5 first, and then Lemma E.4 repeatedly for the remainder of the variables in $S$. This proves the lemma. Note that we need to apply Lemma E.5 several times to a single variable $X \in S$. Indeed we can write

$(Z_i Z_j - 1)\left|(\theta_{k'}^{\|})^T u_{k'} u_{k'}^T \mu_{k'}\right|\left|\mu_k^T u_k(\theta_k^{\|})^T u_k\right| = (\mathbb{E}_\ell \exp(|t_\ell X|)S_\ell - 1)B|X|^c$

$= (\mathbb{E}_\ell S_\ell(\exp(|t_\ell X|) - 1))B|X|^c + (\mathbb{E}_\ell S_\ell - 1))B|X|^c$

for some distribution on $\ell$, and for some terms $S_\ell, t_\ell$, and $B$ that are independent of $X$, and $c \in \{0, 1, 2\}$. Then to take the expectation of this term over $X$, we first apply Lemma E.5 to on $X$ to the first term, and iteratively apply Lemma E.5 to the random variables appearing in the next terms. $\square$

*Proof of Lemma E.11.*

$\frac{1}{1 - x^2}\mathbb{E}_S\left[p_{ii}p_{ij}\theta_k^T h_1(S)\right] = \mathbb{E}_S\left[p_{ii}p_{ij}\left((\theta_{k'}^{\|})^T u_{k'} u_{k'}^T \mu_{k'}\right)\left(2(\theta_k^{\|})^T u_k \alpha_k(\theta_k^{\|})^T u_k\right)\right]$

$= \mathbb{E}_S\left[q_{ii}q_{ij}\left((\theta_{k'}^{\|})^T u_{k'} u_{k'}^T \mu_{k'}\right)\left(2\alpha_k((\theta_k^{\|})^T u_k)^2\right)\right]$

$+ \mathbb{E}_S\left[(p_{ii}p_{ij} - q_{ii}q_{ij})\left((\theta_{k'}^{\|})^T u_{k'} u_{k'}^T \mu_{k'}\right)\left(2\alpha_k((\theta_k^{\|})^T u_k)^2\right)\right]$

$= 2\alpha_k q_{ii}q_{ij}\theta_{k'}^T \mu_{k'}\|\theta_k^{\|}\|^2 + 2\alpha_k q_{ii}q_{ij}\mathbb{E}_S\left[\left(\frac{p_{ii}p_{ij}}{q_{ii}q_{ij}} - 1\right)\left((\theta_{k'}^{\|})^T u_{k'} u_{k'}^T \mu_{k'}\right)\left((\theta_k^{\|})^T u_k\right)^2\right].$

Now by Claim E.16, we have $\left| \frac{p_{ii}p_{ij}}{q_{ii}q_{ij}} - 1 \right| \le Z_i Z_j - 1$, so

$$\left| \mathbb{E}_S \left[ \left( \frac{p_{ii}p_{ij}}{q_{ii}q_{ij}} - 1 \right) \left( (\theta_{k'}^{\parallel})^T u_{k'} u_{k'}^T \mu_{k'} \right) \left( (\theta_k^{\parallel})^T u_k \right)^2 \right] \right| \le \mathbb{E}_S \left[ (Z_i Z_j - 1) \left| (\theta_{k'}^{\parallel})^T u_{k'} u_{k'}^T \mu_{k'} \right| \left( (\theta_k^{\parallel})^T u_k \right)^2 \right]$$

$$\le C \left( \|\theta_k\|^2 + \|\theta_{k'}\|^2 \right) \|\theta_{k'}^{\parallel}\| \|\theta_k^{\parallel}\|^2,$$

Again the second inequality follows from applying Lemma E.5 first (several times as described in the previous lemma), and then Lemma E.4 repeatedly for the remainder of the variables in $S$. Taking absolute values proves the lemma.

$\square$

