# OpenReview forum: "The Dark Side of Invariance: Revisiting the Role of Augmentations in Contrastive Learning"
_ICLR.cc/2023/Conference — Submitted to ICLR 2023_

### Official Review · Reviewer_AVvr · 2022-10-19

**Confidence:** 3
**Correctness:** 2
**Technical Novelty And Significance:** 2
**Empirical Novelty And Significance:** 2
**Recommendation:** 5

**Clarity, Quality, Novelty And Reproducibility:**

As I wrote above, clarity is high except the theory part (Section 4).

I suspect the quality of Theorem 1. The empirical part looks fine.

I feel the novelty is limited (see Weakness 1).

When the code becomes publicly available, the reproducibility should be OK.

**Strength And Weaknesses:**

Strengths
1. Overall the paper is well written.
1. Motivation is clear and the problem this paper addresses is important.

Weaknesses
1. The novelty is somehow limited. I recognized that the main contributions are two-fold: (a) Empirically finding that label-disturbing augmentation is useful, contrary to common opinion. (b) Theoretical analysis of that. Although both are nice, this paper doesn't provide a concrete outcome (new method, new algorithm, etc).
1. I have several concerns for the theoretical part.
- The problem considered in 4.1, to estimate the correlation between u and v can be analytically solved by the eigenvalue decomposition of the covariance $cov[u, v]=E[uv^\top]$. For simplicity, let K=1. The empirical covariance $\Sigma = 1/n \sum_i u_i v_i^\top $ is then converges to $\alpha \mu\mu^\top$ for large n, and $\mu$ is given as the largest eigenvector of $\Sigma$ where $\alpha$ is the eigenvalue. Similarly, the correlation btw u and $\tilde{v}$ is given as the eigenvector of $cov[u, \tilde{v}]$. In this case, the eigenvector is the same $\mu$, but the eigenvalue is decreased from $\alpha$ to $\beta\alpha$ ($\beta \le 1$). The scale of eigenvalue determines the strength of true signal (i.e. correlation) and larger eigenvalue makes the estimation problem easier. So the noisy problem ($cov[u, \tilde{v}]$) is more difficult to obtain an accurate solution. This observation contradicts what Theorem 4.1 suggests. Why does this happen? Did I overlook something?
  - Note that for $K>1$ we can have the same idea by thinking $K$ eigenvalue decomposition problems.
- As mentioned above, the noisy problem should be vulnerable to data noise. However, the speech results (table 2) show the radical augmentation makes NNs robust to noise. Why this is possible?
- Although the theoretical part is intriguing, it is written a bit dense. Also, any intuitions behind Theorem 4.1 are not explained. Since the formal proof is over 10 pages, I really want some proof sketch in the main paper.
- It's better to contain experiments that verify the theorem.

**Summary Of The Paper:**

In this paper, the role of data augmentation in contrastive learning is studied, especially in terms of whether label-preserving augmentation is useful or not. First the authors investigated the empirical evidence using a Viewmaker network (Tamkin et al., 2021b), which automatically finds effective augmentation. The results show that the augmentation that changes label information makes NNs more robust and generalizable to many downstream tasks. To explain this, a theoretical analysis is presented.

**Summary Of The Review:**

While the motivation is interesting, the novelty is not significant enough. Also, the theoretical analysis is counterintuitive and needs more careful assessment.

---

> ### Author Response · Authors · 2022-11-14
> **Response**
>
> We thank the reviewer for their review! We are glad they found that our paper has a clear motivation and addresses an important problem.
>
> ### Contributions do not include new algorithm or method
>
> While we definitely appreciate that some conferences prioritize methods papers, we note that the ICLR 2023 Reviewer Guide [3] explicitly welcomes other kinds of contributions, including conceptual and theoretical contributions. The guide stresses that a good paper "does not necessarily require state-of-the-art results" but instead can "draw attention to an application or problem" or "introduce and/or explain a new theoretical finding".
>
> It is true that our work does not attempt to build a better contrastive learning algorithm; instead, we aim to clarify **why** certain algorithms are better than others, questioning the established consensus in the field. We believe this has inherent scientific interest, and also has the potential to result in better methods that are able to more effectively balance competing features in self-supervised learning.
>
> [3] https://iclr.cc/Conferences/2023/ReviewerGuide
>
> ### Questions about the theory
> **Correlation model contradicts theorem?** The reviewer's understanding of the data setting is correct, however the setting of our theorem is different: it concerns how adding noise in one feature improves the learning of the other features. The theorem involves adding noise to the $k'$th feature, and proves that the kth feature will then be learned better, for any $k \neq k'$. Thus there must be more than one feature (K > 1) for our theorem to apply.
>
> **Robustness to noise** Table 2 does not investigate robustness to data noise; the task is to classify the type of background noise in the input (e.g. cafe, machinery, traffic). Here, the Viewmaker Network is able to capture both features in the input (the speech and background sound) well, while the expert augmentations experience a dramatic decline. We have edited the paper to refer to these as "background sounds" sounds instead to avoid confusion with the other kinds of noise we refer to in the paper. We thank the reviewer for raising this point, as we believe it improves the clarity of the paper.
>
> **Proof Sketch.** We have revised our manuscript to include a brief intuition of the proof (as below) at the beginning of Section E.3 (formerly Section D.3, see "Intuition for proof of Theorem 4.1."). We appreciate this suggestion, and believe it a nice complement to the technical overview of the proof approach given in this section.
>
> > Our proof involves comparing the gradient of the loss in the $\theta_k$ direction, $\nabla_k := \frac{\partial}{\partial \theta_k} \mathcal{L}$ in the setting with noise to the setting without noise. Loosely, our goal is to show that for any $k$, the projection of the this gradient onto the ground truth direction, $\mu_k^T\nabla_k \text{sign}(\mu_k^T\theta_k)$, increases when when increase the noise. The main intuition comes from an expansion of this gradient in Lemma E.7, which shows that $\mathbb{E}\mu_k^T\nabla_k \text{sign}(\mu_k^T\theta_k)$ approximately scales with $\sum_{i}(1 - p_{ii})$. Now observe that $p_{ii}$, the probability of correctly matching the $i$th view to its pair, decreases when we add noise to feature $k'$. Thus adding noise will increase $\mu_k^T\nabla_k \text{sign}(\mu_k^T\theta_k)$, thereby improving the alignment.
>
> **Request for Simulations.** We have performed end-to-end simulations for the full trajectory of gradient descent in the setting of our theoretical result to support our theorem, and have revised our manuscript to include them in the new Appendix D. In brief: we compare contrastive learning with and without adding noise on data generated from the model of Section 4. We consider data with one weak feature (correlation $\alpha_1 < 0.5$), and 50 dominant features ($\alpha_k = 1$). In one simulation, we run 150 iterations of GD, and in the second simulation, we add noise ($\alpha_k \rightarrow 0.25$) to the dominant features after 50 iterations of GD. We observe a significant improvement for learning the weak feature in the setting with added noise. With $\alpha_1 \in  [0.125, 0.25, 0.375, 0.5]$ respectively, the alignment $\frac{\mu^T\theta}{\|\theta\|}$ of the weak feature after 150 iterations goes from $[0.55, 0.67, 0.8, 0.9]$ with no noise to $[0.75, 0.93, 0.98, 0.99]$ with noise. We additionally observe that the alignment of dominant features only decreases slightly (< 0.03) when we add noise. We thank the reviewer for the suggestion, as we feel these simulations provide additional evidence for the phenomena we explore.

---

### Official Review · Reviewer_WzK9 · 2022-10-23

**Confidence:** 4
**Correctness:** 2
**Technical Novelty And Significance:** 4
**Empirical Novelty And Significance:** 3
**Recommendation:** 5

**Clarity, Quality, Novelty And Reproducibility:**

The message and experimental findings about ViewMaker networks are original, to the best of my knowledge. The paper is clearly written and easy to follow for most part.

**Strength And Weaknesses:**

**Strengths**

- This paper raises interesting points about how label-destroying augmentations may not be harmful for contrastive learning, questioning the conventional wisdom about contrastive learning and the role of augmentations.

- The experimental findings on the hybrid datasets that ViewMaker networks learn more label-destroying augmentations than “standard” augmentations but perform slightly (for CIFAR) or much better (for audio), are also very interesting.

- The paper is clearly written and easy to follow for most part

**Weaknesses**

My main concern is the claim from the abstract that “label-destroying augmentations are often crucial in the foundation model setting” is not adequately justified. While the paper sufficiently argues why label-destroying augmentations *may not be harmful*, the results in the paper fail to convince me that they are *crucial*.

(W1) The main evidence provided is the ViewMaker experiments, but in my view the findings are not sufficient to conclude that label-destroying augmentations from ViewMaker were “crucial” for its superior performance.
There could be many other reasons why those are better augmentations and it is also possible (in theory) that the label destroying augmentations are somehow even holding it back. I believe that some more experiments, where the only thing that changes is the level/amount of label-destroying augmentations, would be required to make such a conclusion.

One idea that comes to mind is the following: Run contrastive learning with only those ViewMaker augmentations that do not destroy labels a lot (e.g. pick augmentations that have P(correct) greater than some threshold in Figure 3). If filtering out such augmentations leads to a decline in performance, that would provides evidence that label destroying augmentations were indeed crucial.
Some such experiment that isolates the effect of the presence or absence of label destroying augmentations, without other potential confounding factors, seems important for the conclusions in the paper.

(W2) Similarly the theoretical result from Section 4 does not characterize the necessity or benefit of label destroying augmentations well enough.
The result shows that if noise is added to some feature (to “destroy” it), then one step of gradient descent will lead to better learning of other features.
This result fails to paint a more global picture of what would happen if contrastive learning is run with augmentations that do and do not destroy augmentations/features every now and then.
While interesting, the one step gradient result that is presented is quite intuitive, since at the extreme if one feature is completely destroyed, then other features start becoming more important for the contrastive loss.
However it is not clear to me that the result provides a “simple linear model that captures the essence of how label-destroying augmentations can improve downstream accuracy “, as claimed on page 6.
Some more comments:
- Is this result saying that with label-destroying augmentations, the useful features will be learned faster? Because it seems to me (although I do not have a proof) that with or without label-destroying augmentations, all the features should be learned eventually. I'm not sure if any suppression of features will happen without the occasional destruction of features.
- Even if an end-to-end result cannot be shown, even some simulation experiments with contrastive learning on this example for two augmentation schemes, one that occasionally destroys features while another that doesn’t destroy, might help shed some more light and strengthen the claim that label destroying augmentations can help.
- Will help to include a proof sketch or at least the key steps in the proof, even if for a special case like \beta=0, for the reader to get some intuition.


Other comments

- In Section 2, I’m not sure if an augmentation “applied repeatedly” is the right way to think of this. A function $f$ might just be invariant on the support of the input distribution and one augmentation applied to those images, and not necessarily on augmentations of augmentations. This will not lead to the “catastrophic augmentation collisions” that is shown in Appendix B.

- The bimodal distribution in probabilities in Figure 2 is quite interesting. However there might be some confounding factors since the histograms for ViewMaker and Expert augmentations use different network (based on the description in Appendix C).
To get rid of confounding factors it would help to plot histogram from both using both networks (so 4 plots instead of 2). Was this image randomly selected? Is the finding true for other images as well?

**Summary Of The Paper:**

This paper questions the widespread argument/intuition for the success of contrastive learning that a good augmentation must be label preserving. The paper argues that label-destroying augmentations may be important for contrastive learning to learn diverse general purpose representations. It further hypothesizes that label destroying augmentations serve as some kind of “feature dropout” since they can prevent one “shortcut feature” from suppressing other features. The following arguments are provided to support the hypothesis:
- Discussion (in Section 2) about why some standard augmentations are not label-preserving and why invariance to those augmentations is undesirable. E.g. while random gray scaling is an augmentation, we often do not want invariance w.r.t. it since color information can be useful for downstream tasks.
- Experiments with ViewMaker networks (a recent method to automatically learn augmentations) on hybrid datasets (e.g. CIFAR with superimposed digits) where the learned augmentations are generally better than standard augmentations. The paper observes that despite performing better, ViewMaker has more label-destroying tendencies, as verified with some visualizations of the augmentations and other statistical evidence (questions about this later)
- A theoretical result with a linear model that tries to capture that idea that adding noise to some features (to simulate label destruction) can speed up the learning of other features (for one gradient step).

**Summary Of The Review:**

The message and experimental findings about ViewMaker networks are original, to the best of my knowledge, and overall I find them quite interesting. However I believe that the paper is lacking in execution of the ideas, more precisely in its justification of the *necessity* of label-destroying augmentations. I am thus inclined to assign a score of weak reject, but I would be happy to change the score if there is some more justification for the point about necessity, through experiments or theory.

---

> ### Author Response · Authors · 2022-11-14
> **Response [1/2]**
>
> We thank the reviewer for their review! We are glad they found the main point of our paper interesting, as well as our experimental findings.
>
> ### W1: Are the label-destroying augmentations necessary?
>
> We thank the reviewer for raising this point. The word "crucial" in the abstract was intended to refer to the theoretical results, where we prove that adding noise improves learning of other features. These are bolstered by our new simulations in Appendix E, which show this effect across full runs of gradient descent. In addition, we provide some additional empirical evidence for this effect in neural networks (detailed below), and we also add a bit more nuance in the abstract  ("can be useful") to reflect the perspective of the rest of the paper.
>
> **Experiment to isolate effect of feature dropout.** We thank the reviewer for the very thoughtful suggestion in W1 to get at this question. We ran a similar experiment to explore the same question: train with the viewmaker, but protect the object feature with a mask so its pixels cannot be affected by the viewmaker. This enables us to see whether the viewmaker's inability to destroy the simpler feature impairs the learning of the CIFAR class. Indeed, we do find that this is the case. CIFAR accuracy declines from 79.8 to 26.0 for the CIFAR+Shape dataset and 69.3 to 50.6 in the CIFAR+Digits dataset, while remaining roughly constant for the shape and digit label respectively. This highlights the importance of label-destroying augmentations. The CIFAR+Letter experiments saw decreases in both features (CIFAR and Letter); we hypothesize this is because the color of the letter suppresses both the CIFAR-10 class and the letter class. We have included these results in the most recent draft in Appendix C.
>
> We believe a variant of the suggestion in W1 may be an interesting direction for future work, providing one has enough GPU memory to run a pretrained evaluation model for filtering alongside the main model, and is able to keep generating views until all examples in the batch have label-preserving viewmaker augmentations.
>
> **Bimodal feature graphs**
>
> We thank the reviewer for the suggestion. We have rerun the experiments, aggregating predictions from augmentations of 1200 different examples from each dataset. We again see no bimodality for the expert views, and clear bimodal distributions when using viewmaker views. We also ran the experiments with the same network and views, though these exhibit less structure because the network has adapted to the characteristics of the views. We have updated the paper with these newest figures (Figures 5 and 6).
>
>
> ### W2: Additional clarifications about our theoretical results
>
> **Will all features be learned eventually?** The reviewer is correct that all the features will be learned eventually. However, the speed at which the features are learned will vary based on the level of feature suppression. By studying one step of gradient descent, we can see how suppressing a feature (that is already partially learned) can speed the learning of the other features.
>
> **End-to-end simulation results.**  We have performed end-to-end simulations in the setting of our theoretical result to support our theorem, and have revised our manuscript to include them in the new Appendix D. In brief: we compare contrastive learning with and without adding noise on data generated from the model of Section 4. We consider data with one weak feature (correlation $\alpha_1 < 0.5$), and 50 dominant features ($\alpha_k = 1$). In one simulation, we run 150 iterations of GD, and in the second simulation, we add noise ($\alpha_k \rightarrow 0.25$) to the dominant features after 50 iterations of GD. We observe a significant improvement learning the weak feature in the setting with added noise. With $\alpha_1 \in  [0.125, 0.25, 0.375, 0.5]$ respectively, the alignment $\frac{\mu^T\theta}{\|\theta\|}$ of the weak feature after 150 iterations goes from $[0.55, 0.67, 0.8, 0.9]$ with no noise to $[0.75, 0.93, 0.98, 0.99]$ with noise. We additionally observe that the alignment of dominant features only decreases slightly (< 0.03) when we add noise.
>
> We agree that theoretically understanding this model more globally (for the entire training process) would be interesting. Because of the challenge analyzing GD on the cross entropy contrastive loss, we leave this for future work.

---

> ### Author Response · Authors · 2022-11-14
> **Response [2/2]**
>
> ### Proof Intuition.
>
> We thank the reviewer for the suggestion. We have revised to include a brief intuition of the proof (as below) at the beginning of Section E.3 (formerly Section D.3). This supplements a detailed technical overview of the proof approach, already given in this section.
>
> > Our proof involves comparing the gradient of the loss in the $\theta_k$ direction, $\nabla_k := \frac{\partial}{\partial \theta_k} \mathcal{L}$ in the setting with noise to the setting without noise. Loosely, our goal is to show that for any $k$, the projection of the this gradient onto the ground truth direction, $\mu_k^T\nabla_k \text{sign}(\mu_k^T\theta_k)$, increases when when increase the noise. The main intuition comes from an expansion of this gradient in Lemma E.7, which shows that $\mathbb{E}\mu_k^T\nabla_k \text{sign}(\mu_k^T\theta_k)$ approximately scales with $\sum_{i}(1 - p_{ii})$. Now observe that $p_{ii}$, the probability of correctly matching the $i$th view to its pair, decreases when we add noise to feature $k'$. Thus adding noise will increase $\mu_k^T\nabla_k \text{sign}(\mu_k^T\theta_k)$, thereby improving the alignment.
>
> ### Augmentations applied repeatedly
>
> It is true that our argument relies on the augmented images having support on data distribution. We believe this is a reasonable assumption for the augmentations we consider. For example, natural images exist with a range of exposures, ranging from very underexposed to overexposed, corresponding to the full spectrum of brightness shift augmentations. Likewise, images have varying degrees of saturation or hue shifts depending on lighting conditions, sensor noise, or other environmental factors.
>
> However, we have also expanded Section 2 to focus on the point that many augmentations target information (e.g. brightness or color) that we do not want to be fully invariant to, which matters even if the augmentation is only applied once. We believe this improves the section, and thank the reviewer for the suggestion.

---

> ### Comment · Reviewer_WzK9 · 2022-11-19
> **Response to authors**
>
> I thank the authors for their response and the new experiments. However I am not convinced that the new experiments fully address my main concern about why label destroying augmentations can help. The more I think about it, the less clear the goal/meaning of “feature dropout” in this paper becomes to me. From the results in the paper and the new experiments the phenomenon of interest seems to be the following:
>
> *Suppose you have 2 features f1 and f2. If f2 is randomly dropped out/destroyed, then this helps learning f1 better.*
>
> I would argue that this phenomenon is not counter-intuitive and actually does not break conventional wisdom. Calling this setting “label destroying” is a bit misleading because the label we care about here corresponds to f1, which is not being destroyed.
>
> In the new experiment in Table 3, the inferior performance of Mask-Viewmaker (where the shape f2 is not destroyed) on C-10 (f1) is not entirely surprising and does not justify the benefit of label destroying augmentations, since the label for f1 is not being destroyed. The inferior performance on the object (f2) could simply be because no augmentations are being applied to the shape f2 at all, thus making contrastive learning ineffective. (I am assuming that "object" in "protect the object feature" refers to the new shape/digit etc. and not the original image).
>
> Similarly, in the new synthetic experiment, dropping some dominant features f2 after a particular iteration will intuitively (but not obviously) make the other feature f1 more important and thus improve its learning. While this is interesting, it is still not breaking conventional wisdom.
>
> The true benefit of label-destroying augmentation through feature dropout would be if occasionally dropping f2 leads to better performance of *both* f1 and f2 (which may be the case for the original ViewMaker experiment) OR if occasionally dropping *both* f1 and f2 can help learning *both* f1 and f2 faster. Experimentally testing this hypothesis for the ViewMaker setting might require some more thought, but I do not think that the current  experiments justify this. This phenomenon is easier to test in the synthetic setting, e.g. in one setting there is no dropping of features, whereas in another setting both features are randomly and occasionally dropped. If the latter setting learns both features faster, that would be a more convincing justification of feature dropout.
>
> Another follow up question: Are ViewMaker augmentations learned specifically for each task (C+shape, C+digit etc.)? This was not clear to me. If true, then this automatically provides ViewMaker an advantage over Expert augmentations because of access to modified images.
>
> Overall, although I was convinced that label destroying augmentations may not be bad, I am still not convinced that they are beneficial. More thought needs to go into defining feature dropout, how to experimentally validate the claims, and these need to be more clearly presented.

---

> > ### Author Response · Authors · 2022-11-23
> > **Response**
> >
> > We thank the reviewer for the great questions and suggestions.
> >
> > **Can the paper's takeaway be summarized as ``Suppose you have 2 features f1 and f2. If f2 is randomly dropped out/destroyed, then this helps learning f1 better''?**
> >
> > This is half of the picture; crucially, we also show that you can also *continue to learn f2 well* even in the presence of f2-destroying augmentations.
> >
> > That is, while these augmentations are typically understood as encoding **invariance** to f2, we show this is not the case; they can still enable learning f2 while enabling learning of (possibly many) f1's much better. This picture is reinforced by our new end-to-end empirical simulations in Figure 6, which show that adding noise to the strong feature can reduce its learning by a negligible amount, while *dramatically* improving the learning of many other features. They are also validated by our new results in Figure 4, aggregated over 1200 examples, which demonstrate that you can learn f2 well even in the presence of augmentations that target f2 a significant fraction of the time.
> >
> > Both of these results are also in stark contrast with existing theoretical models of contrastive learning which require f2 label-destroying augmentations to *never* occur, or occur an *``extremely small''* fraction of the time (§5, para. 1). Thus, our work identifies a gap between existing theoretical understandings of contrastive learning and empirical reality.
> >
> >
> > **Can dropping f1 and f2 lead to better learning on both f1 and f2?**
> >
> > In the scope of our paper, which focuses on different features, we do not expect adding noise to f2 will help f2 be learned better. That said, an exciting direction for future work is to consider performance on *tasks* that depend on multiple features. We can sketch out one conceptual argument where the insights in our paper can shed light on this question:
> >
> > Consider a distribution of images with four features: f1, f2, f3, f4. Features f1 and f2 are both needed for task t1, while f3 and f4 are both needed for task t2. Assume f1 is learned well and properly captured 100% of the time by the network, but it suppresses features f2, f3, f4 so they are learned 0% of the time. This would result in very poor performance on downstream tasks t1 and t2.  Now assume we drop out f1, and it is now only learned 80% of the time, but features f2, f3, f4 are now learned 60% of the time. Now the performance on tasks t1 and t2 will be much better than before, even though all we have done is add noise to f1.
> >
> > The key difference between this setting and the one we explore in our paper is the nonequivalence between features and tasks, which gives rise to this phenomenon. We agree a discussion of this future direction would strengthen our work and thank the reviewer for the suggestion
> >
> > **Does the viewmaker have an advantage over the expert views?**
> >
> > The viewmaker and the expert views both receive the same input images, and neither receives a mask or additional side information specifying the suppressing features. The difference is that viewmaker is an adaptive method that learns augmentations automatically as part of the pretraining process [1]. This enables it to identify the different features in the input and learn suitable augmentations (e.g., Figure 1) without the need for expert trial and error to identify and drop out the suppressing features.
> >
> > We thank the reviewer for again the great questions and suggestions. Please let us know of any remaining concerns.
> >
> > [1] Viewmaker Networks: Learning Views for Unsupervised Representation Learning, Alex Tamkin, Mike Wu, Noah Goodman

---

### Official Review · Reviewer_mMWc · 2022-10-24

**Confidence:** 4
**Correctness:** 3
**Technical Novelty And Significance:** 3
**Empirical Novelty And Significance:** 3
**Recommendation:** 6

**Clarity, Quality, Novelty And Reproducibility:**

Overall the paper is clear and easy to follow. I have the following suggestions to make about the clarity of the paper:

- I believe that expanding upon the motivating example in Section 2, as stated above, is useful in order to improve the reader’s understanding of the motivation behind the paper.
- Regarding the theoretical result, I believe the explanation on the addition of the noise before Theorem 4.1 should be included in the Theorem itself, in order for the expression of the corrupted feature to be clearer to the reader.

With respect to the novelty, as stated above I believe that the paper provides a novel interpretation on the necessity of invariance across augmentations. I consider this interpretation to be original.

The authors provide details in order to reproduce their experiments, and plan to release their codebase.

**Strength And Weaknesses:**

Strengths:
- This is an interesting paper, in that it questions a commonly accepted paradigm for self supervised learning, in that the augmentations applied on the inputs need to be carefully selected, so as not to affect the downstream task. Since this field is overall novel, I believe that challenging commonly accepted views is important for the field as a whole.

- The authors also make a convincing argument regarding the use of feature destroying augmentations, in the setting of multitask learning. According to the authors, while previous works only consider the setting of a single downstream task, they argue that in order for a model to perform well in a multitude of different downstream tasks, no single feature should be prioritized over the others. This is an interesting argument for the case of the use of a variety of augmentations, whether they are label preserving or not.

- The experiments provided by the authors demonstrate that the use of Viewmaker networks to automatically extract (possibly label destroying) views show benefits over the use of expert augmentations, which is an argument for their case that invariance is not necessarily required.

Weaknesses:
- I believe that the motivating examples behind the arguments made by the authors in Section 2 can be improved. In particular, the authors argue that the brightness augmentation used in practice is not label preserving, while this may only be true in extreme cases. Furthermore, I also believe that their motivating argument would be cleaner if they described hue shifts and greyscaling the same way they describe brightness augmentations, in that they may not be label preserving despite their widespread use. I also think that the fact that an augmentation may be invariant for a given task and not for a different one should be stressed here.

- The authors use Viewmaker networks to automatically extract augmentations, and make their central observation that augmentations need not be label preserving based on this. While this is a good argument, I am not sure if it is adequately supported. From the qualitative side, Figure 1 claims that Viewmaker views tend to destroy labels for one of the tasks. A similar argument can be made for the expert views in some cases (particularly in row 4, where the expert views corrupt the input greatly). From the quantitative side, the authors analyze how the predictive capabilities of a classifier are affected by the Viewmaker and the expert augmentations. However, this is done using a single sample of the data, and deriving several views from it. I believe that this should be done with a multitude of the samples of the dataset, in order to avoid any possible bias arising from this sample.

- The authors provide a good theoretical argument on the use of noise on a given feature, and how doing so actually improves the learning of other features. However, in the setting that the authors describe, I find it slightly weird that neither of the two views of the samples that we consider contain by themselves any information about the directions $\mu_k$ that we want to learn. This information is rather incorporated into their joint distribution, and thus it seems more natural to me to consider them both as a single sample, instead of two views of the same object. This makes the theoretical example somewhat disjoint from the rest of the work, if I understand correctly. Moreover, the theoretical result states that if a feature is corrupted, then we can learn the rest of the features well. I think it is also useful to know in the main theorem how much we need to corrupt that feature to achieve this (thus, what is the total effect on the alignment of our learned vectors with the ground truth).


**Summary Of The Paper:**

This paper challenges the common belief in literature in self supervision that augmentations need to be label preserving, in order to perform well in downstream tasks. The authors first show that when using a Viewmaker network to automatically derive good views, the augmentations derived by the model may drop significant information about the label of the downstream task. Using this observation, the authors then demonstrate theoretically that in a simple problem, adding noise to the features of the inputs helps the optimization process to find a better representation.

**Summary Of The Review:**

Overall, I think this is an interesting paper, due to the fact that it presents a novel interpretation on the necessity of invariance in the augmentations. I lean towards acceptance due to the interesting aspects of the above, however I also believe that there are a few issues in the paper that need to be addressed, with respect to how the quantitative evaluation of invariance is performed and how the theoretical result is linked to the rest of the paper.

---

> ### Author Response · Authors · 2022-11-14
> **Response**
>
> We thank the reviewer for their review! We are glad they found our work novel and convincing, and that challenging the commonly accepted paradigm in contrastive learning is important for the field as a whole.
>
> **Improving the explanation of motivating examples in Section 2**
>
> We thank the reviewer for the suggestions. In response, we've made some changes to Section 2, including describing hue shifts and greyscaling in the same way as brightness shifts, and stressing that an augmentation may be invariant for a given task and not for a different one.
>
> **Evaluating feature dropout on multiple samples**
>
> We thank the reviewer for the suggestion. We have rerun the experiments, aggregating predictions from augmentations of 1200 different examples from each dataset. We again see no bimodality for the expert views, and clear bimodal distributions when using viewmaker views. We have updated the paper with these newest figures.
>
> **In the theory section, is it strange that the views by themselves do not contain information about $\mu_k$?**
>
> We think this is fairly similar to the case of images in computer vision: the images themselves do not contain information about the decision boundary between classes (e.g. poodle vs shiba inu, dark dog vs light dog), but once the decision boundary is known you can classify the input. However, the decision boundaries in our case are certainly simpler, being linear, as opposed to the highly nonlinear decision boundaries neural networks find.
>
> **Can the theorem tell us how much corrupting one feature helps the other features?**
>
> Yes, indeed we can glean an estimate from the proof for how much it helps in expectation for one step of GD with step size 1. If we change the correlation of feature $k’$ from $\alpha$ to $\alpha - x$, we achieve a difference
>
> $E_{\text{no noise}}[\text{arcos for kth feature}] - E_{\text{noise}}[\text{arcos for kth feature}] \propto $ $x  \eta  \alpha_k  |\theta_{k'}^T\mu_{k’}|^2*|\theta_k^T\mu_k|/(\sqrt{\|\theta_k\|^2 - (\mu_k^T\theta_k)^2})$
>
> Parsing this, the improvement in the angle of the kth feature from adding noise:
> (a) scales linearly with how much you decrease $\alpha_{k'}$ by.
> (b) Is greater the more $\theta_{k'}$ is oriented in the correct direction.
> (c ) Is greater when $\theta_k$ is partially in the correct direction, but not too close to being aligned.
>
> **"The explanation on the addition of the noise before Theorem 4.1 should be included in the Theorem itself, in order for the expression of the corrupted feature to be clearer to the reader."**
>
> We thank the reviewer for their suggestion. We have made the change in the uploaded revised version.

---

> > ### Comment · Reviewer_mMWc · 2022-11-19
> > **Thank you for your comment.**
> >
> > Thank you for your detailed responses to my comments.
> >
> > I have one final question/clarification based on the overall responses. Regarding the overall learning of features (also mentioned by Reviewer WzK9), it is mentioned that at the end all features will be learned. I would be grateful if the authors were able to further comment on this or possibly quantify it, given that my question on how much we need to corrupt a feature is related. For example, it would be great if there was a simple criterion that decreases in each step of SGD (taking into account all features, not just the ones without the added noise), and which shows that all features will be learnt eventually. I understand that this is difficult in general (as mentioned in the response to Reviewer WzK9), and the end-to-end simulations support this experimentally, but I wonder if a simple case (for example, for a small number $K$ of features) would be feasible to quantify.

---

> > > ### Author Response · Authors · 2022-11-23
> > > **Response**
> > >
> > > Thanks for the great question and suggestions. We can show that all features will be learned eventually arbitrarily well if we regularize such that $\|\Theta\|_2$ is sufficiently small. This is because so long at $\|\theta_k\|_2$ is small enough, and the correlation $\alpha_k > 0$, the expected gradient will increase the alignment of the feature $\theta_k$.
> > >
> > > In more detail, we observe theoretically that there is a ceiling in alignment that depends on the correlation ($\alpha_k$) and the norm of $\theta_k$: up to constant factors, we can show that the angle between $\theta_k$ and $\alpha_k$ will be at most $\frac{\|\theta_k\|^2}{\alpha_k}$. Observe that this angle can be arbitrarily small if we force $\|\theta_k\|$ to be small via l2 regularization. Further, this quantity increases with the noise we add to the $k$th feature, which decreases $\alpha_k$.
> > >
> > > We will add this discussion to the paper, and thank the reviewer for the suggestion. We also share some additional theoretical findings below that further flesh out this picture:

---

> > > > ### Author Response · Authors · 2022-11-23
> > > > **Response (cont.)**
> > > >
> > > > To understand more technically the tradeoffs of adding $\beta$ noise to a dominant feature $k’$, we can also show the following under the assumptions of our theorem:
> > > > 1. For a different feature $k$ (which we don’t add noise to), the increase in alignment of the $k$th feature will be approximately proportional to $\beta \alpha_k*|\theta_{k'}^T\mu_{k’}|^2*|\theta_k^T\mu_k|/\|\theta_k\|*(\sum_i p_{ii})/m$). Note: this is a slight modification of the expression we gave in our original response; to make this simplification, we have assumed that the $k$th feature is not *too aligned* yet. We also included the exact dependence on $P(correct) = \sum_i p_{ii})/m$ to better understand the tradeoff (see below).
> > > > 2. For the feature $k’$ which we add noise to, there are two possibilities.
> > > >       a) The alignment is already extremely good (near its ceiling, as described above), but by adding noise, we decrease the alignment ceiling.
> > > >       b) We are still learning this feature (the alignment is not yet at its ceiling), so adding noise *slows* the rate of learning. In particular, the decrease in alignment of the $k’$th feature (relative to without noise) will be approximately proportional to $\beta  *\phi *\|\theta_{k’}\|^2$, where $\phi$ is the angle between $\theta_{k’}$ and $\mu_{k’}$.
> > > >
> > > >
> > > > We can draw the following conclusions:
> > > > 1. If our objective is to maximize the _minimum_ of the feature alignment over all K features, then (under the assumptions of our theorem), it is always advantageous to add noise to the features that have been learned better than the feature that is least aligned.
> > > > 2. If our objective is to maximize the _average_ feature alignment over all K features, by observing the expressions given in (1) and (2b) above, we can see that adding noise will be advantageous if:
> > > >     -  We have learned the $k’$th feature sufficiently well such that $\phi$ is very small.
> > > >     - $\sum_i p_{ii}$ is large enough, that is, the contrastive model is predicting the correct view with reasonable probability.  This is intuitively similar to the CE loss, $\sum_i-\log(p_{ii})$, not being too large.
> > > >
> > > >     If adding noise to one feature is advantageous, then adding any amount of noise to that feature will be advantageous for one step of SGD. However, if we train for a long time with too much noise, we may decrease the alignment of $k’$ so much that adding noise is no longer good. In practice, one can check that P(correct) =  $\sum_i p_{ii}/m$ does not get too close to $0$, otherwise, one should stop adding noise.
> > > >
> > > >
> > > > Finally, we remark that we observed in our synthetic experiments that the quantity of noise added to the dominant features did noticeably affect the learning of both the dominant and the weak features. In the experiments we show, we added noise such that the correlation of the dominant feature was $0.25$. If we instead eliminated all signal by changing the correlation to $0$, the loss exploded (and $\sum_i p_{ii}$ got small), and the dominant feature was forgotten. When we changed the correlation to $0.5$, the weak feature was not learned as quickly. We will include a discussion of this with the synthetic experiments in a revision.

---

### Official Review · Reviewer_Xou4 · 2022-10-24

**Confidence:** 3
**Correctness:** 4
**Technical Novelty And Significance:** 3
**Empirical Novelty And Significance:** 3
**Recommendation:** 8

**Clarity, Quality, Novelty And Reproducibility:**

The paper is well written, with careful experiments and a relevant theorem (with a very detailed proof in supplement).



**Strength And Weaknesses:**

Strengths:
Rather convincing experiments are presented supporting the hypothesis that data augmentation leads to more balanced feature weights.

Weaknesses: While carefully supported by creative designed experiments the idea is not entirely new. In fact work by Ericsson et al. [1] has pointed to a quite similar conclusion - this work is not cited in the present contribution.

[1] Ericsson, L., Gouk, H. and Hospedales, T.M., 2021. Why do self-supervised models transfer? investigating the impact of invariance on downstream tasks. arXiv preprint arXiv:2111.11398.

The paper's theorem is relatively weak evidence (many conditions, weak conclusion)


**Summary Of The Paper:**

The paper addresses the question, what role does data augmentation play in (contrastive) self-supervised learning? As the authors write they complicate the simplistic view that data augmentation implements invariances: "We complicate this picture by showing that label-destroying augmentations are often crucial in the foundation model setting, where the goal is to learn diverse, general-purpose representations for multiple downstream tasks."

Rather the authors show that data augmentation can lead to a more balanced representation of features (essentially by feature level dropout), so that a broader set of down stream tasks is served.

The evidence presented is experimental, indicating that a combined (ensemble) approach of augmentation interventions lead to better general down stream generalizability. A theoretical analysis of SGD with data augmentation (added noise) complements the experimental evidence.


**Summary Of The Review:**

Clear hypothesis with strong experimental evidence, supported by somewhat weaker theory.

In rebuttal please related the present hypothesis to the prior work of Ericsson et al.

---

> ### Author Response · Authors · 2022-11-14
> **Response**
>
> We thank the reviewer for their review! We are glad they found our work to have a clear hypothesis with strong and convincing experiments that are complemented by the theoretical results
>
> **Relation to Ericsson et al [1]**
>
> We thank the reviewer for sharing this work! We believe the questions it investigates are interesting and connected to our work, albeit with a distinct focus. We have added it to our paper's related work section.
>
> The main difference in focus is that Ericsson et al [1] explores augmentations through the perspective of invariances: depending on the chosen invariance, augmentations can help some tasks and hurt others. For example, augmentations thought to enforce a spatial invariance may aid object recognition tasks but harm pose identification tasks. By contrast, our work explores a different interpretation for augmentations as a form of feature dropout. We show how label-destroying augmentations, which sometimes destroy features of interest, can actually serve a beneficial role: networks can still learn the partially-destroyed features while learning other features better.
>
> Another difference is that Ericsson et al [1] explores a setting where the two kinds of features (spatial and appearance) are known in advance. This enables Ericsson et al [1] to achieve good performance by training two models, one with each type of augmentation, and concatenating the resulting representations together. In our work, we show that it is possible to achieve balance between features in a single model without this knowledge (e.g. through the use of Viewmaker networks).
>
> **Number of conditions in theorem**
>
> Although our theorem does have a few conditions, we think these conditions are quite reasonable and intuitive (see also the end of Section 4.2):
> - **Assumption 1** says that the theorem applies to features which have not yet been learned already, which is intuitive.
> - **Assumption 2** states that the feature to which we are adding noise has been learned at least a small amount already. This is reasonable because the easiest features to learn are the ones we will benefit from adding noise to.
> - **Assumption 3** holds as long as the weights of the model are not too large, which would destabilize training. This is likely to hold in practice given that contrastive networks are initialized with small weights and commonly trained with weight decay (e.g., as in SimCLR [1] and MoCo [2]). Note that our theoretical results also apply when weight decay is used.
>
> [1] A Simple Framework for Contrastive Learning of Visual Representations, Chen et al, 2020
> [2] Momentum Contrast for Unsupervised Visual Representation Learning, He et al, 2020

---

### Author Response · Authors · 2022-11-14
**General Response**

We thank the reviewers for their reviews! We have responded to the main comments beneath each review, and also uploaded a new version incorporating these changes. Please do not hesitate to comment with additional questions—we are happy to address any remaining concerns.

---

### Decision · Program_Chairs · 2023-01-20

**Decision:**

Reject

**Justification For Why Not Higher Score:**

See above

**Justification For Why Not Lower Score:**

N/A

**Metareview: Summary, Strengths And Weaknesses:**

- Summary:

This paper challenges the common belief in self supervision that augmentations need to be label preserving, in order to perform well in downstream tasks. The authors first show that when using a Viewmaker network to automatically derive good views, the augmentations derived by the model may drop significant information about the label of the downstream task. Using this observation, the authors then demonstrate theoretically that in a simple problem, adding noise to the features of the inputs helps the optimization process to find a better representation.

- Strengths:

1. Interesting topic/open question; all reviewers acknowledge that this is a topic that we should focus on more.
2. The paper is clearly written, and (at least based on the experiments provided) the arguments are supported empirically.

- Weaknesses:

1. Several reviewers raised the concern regarding how the quantitative evaluation of invariance is performed and how the theoretical result is linked to the rest of the paper. The discussion with the reviewers was rather extensive; however, not all concerns where either addressed or could be added to the paper without major revision.
2. Even after discussion, one reviewer still believes that "More thought needs to go into defining feature dropout, how to experimentally validate the claims, and these need to be more clearly presented." We encourage the authors to consider the suggestions made by the reviewers holistically when revising the paper.

- Recommendation:

First of all, the authors should be stayed assured that there were discussions regarding the paper, and the area chair read carefully i) the paper, ii) the reviews, and iii) the discussions after the rebuttal. Also, reviewers definitely agree with the authors that any lack of algorithmic contributions should not necessarily be a factor for rejection; yet, the authors should acknowledge that in these case, a strong and thorough experimental validation and verification of the proposed arguments and hypotheses should support the paper. We acknowledge that the revised paper addresses concerns raised, but the reviewers have remaining concerns.

We recommend the authors include a more thorough experimental validation for their methodology proposal: including experiments that are common in various fields should be sufficient to answer any questions about the empirical validation of the method. As an area chair, I found the concerns raised by reviewers important to be handled, but I would also consider this case a fortunate one, since this is definitely a constructive feedback to your paper.

We recommend the authors to consider these (common over all reviewers) concerns to revise their paper; we strongly support for a resubmission to a near-future ML conference.